# Model cascade from meteorological drivers to river flood hazard: flood-cascade v1.0

Peter Uhe[1,2], Daniel Mitchell[1,2], Paul D. Bates[1,2], Nans Addor[3,4], Jeff Neal[1,2], and Hylke E. Beck[5]

[1]Cabot Institute for the Environment, University of Bristol, Bristol, UK
[2]School of Geographical Sciences, University of Bristol, Bristol, UK
[3]Climatic Research Unit, School of Environmental Sciences, University of East Anglia, Norwich, UK
[4]Geography, College of Life and Environmental Sciences, University of Exeter, Exeter, UK
[5]Department of Civil and Environmental Engineering, Princeton University, Princeton, USA

**Correspondence:** Peter Uhe (pete.uhe@gmail.com)

**Abstract.** Riverine flood hazard is the consequence of meteorological drivers, primarily precipitation, hydrological processes and the interaction of floodwaters with the floodplain landscape. Modeling this can be particularly challenging because of the multiple steps and differing spatial scales involved in the varying processes. As the climate modeling community increases their focus on the risks associated with climate change, it is important to translate the meteorological drivers into relevant hazard estimates. This is especially important for the climate attribution and climate projection communities. Current climate change assessments of flood risk typically neglect key processes, and instead of explicitly modeling flood inundation, they commonly use precipitation or river flow as proxies for flood hazard. This is due to the complexity and uncertainties of model cascades and the computational cost of flood inundation modeling. Here, we lay out a clear methodology for taking meteorological drivers, e.g., from observations or climate models, through to high-resolution (~90 m) river flooding (fluvial) hazards. Thus, this framework is designed to be an accessible, computationally efficient tool using freely available data, to enable greater uptake of this type of modeling.

The meteorological inputs (precipitation and air temperature) are transformed through a series of modeling steps to yield, in turn, surface runoff, river flow, and flood inundation. We explore uncertainties at different modeling steps. The flood inundation estimates can then be related to impacts felt at community and household levels to determine exposure and risks from flood events. The approach uses global data-sets and thus can be applied anywhere in the world, but we use the Brahmaputra river in Bangladesh as a case study in order to demonstrate the necessary steps in our hazard framework. This framework is designed to be driven by meteorology from observational data-sets or climate model output. In this study, only observations are used to drive the models, so climate changes are not assessed. However, by comparing current and future simulated climates, this framework can also be used to assess impacts of climate change.

## 1 Introduction

Flooding is a natural phenomena which is a part of the lives of many people around the world. Flood waters can be important in depositing nutrients and encouraging plant growth on floodplains (Johnston et al., 1984; Ogden and Thoms, 2002). Conversely,

they are also hugely damaging, causing thousands of lives lost and billions of dollars of damage (Wallemacq and House, 2018; Guha-Sapir et al., 2016). Most inland floods are driven by precipitation events but culminate from the interaction of factors such as soil-moisture, evaporation, river channel routing and floodplain dynamics. It is these non-linear interactions which determine the ultimate magnitude of the resulting flood hazard (Sharma et al., 2018; Grimaldi et al., 2019). The flood impacts are furthermore the result of the exposure and vulnerability of populations.

The hydrological cycle is being altered due to the influences of climate change. So making flood inundation models more accessible for use in climate change impacts studies is an important step for future research. The approach presented here is designed to enable more robust flood inundation estimates using climate model outputs, over larger regions. This study focuses on describing the model framework and evaluating its performance against past observations of river flow and flood inundation. The model framework is compatible with climate model outputs, which use the same data format as gridded meteorological data. However, we do not include climate projections here, which will be covered in a future study.

The use of modeling cascades to produce flood hazard estimates is not a new concept. However, there are still limited number of studies designed to investigate climate change using high resolution (< 1km) flood inundation models. Studies which have combined climate change projections with high resolution flood inundation modeling have been small-scale case studies (e.g., Ranger et al., 2011; Schaller et al., 2016; Hattermann et al., 2018), or used global flood hazard models (e.g., UNISDR, 2015; Alfieri et al., 2017; Winsemius et al., 2013, 2016; Hirabayashi et al., 2013; Ikeuchi et al., 2015). We note that these global flood hazard models have limitations, and (with the exception of Alfieri et al. (2017)), flood hazard is simulated at coarser resolution (10–30 km), then downscaled to resolutions of 1 km or less, rather than hydraulically simulating flooding at the smaller scales. Hence these coarser models are not able to represent the smaller scale floodplain dynamics and connectivity.

Relevant to the goal of regional scale flood modeling at high resolution, large scale hydrological models have also been coupled to flood inundation models and these also have the capability of being driven by climate model outputs. For example, model cascades have been run over the Ohio river basin in the USA (Rajib et al., 2020), the Elbe river basin in Germany (Falter et al., 2016) and the Murray Darling basin in Australia (Grimaldi et al., 2019). One key thing to note with these approaches is that they used large numbers of river discharge stations for the calibration of their hydrological models. This is a successful approach where good quality observations are available, however this is not as easily applicable in poorly gauged regions or for global studies. In those situations, parameter regionalization, where model parameters determined in catchments with high-quality observational data are used to model other catchments which have similar characteristics (Kokkonen et al., 2003; Beck et al., 2016) may be more appropriate. Another model framework, GLOFRIM (Hoch et al., 2017, 2019), approaches this problem by driving high resolution flood inundation models, run over small regions, by global hydrological model simulations. The framework presented here uses similar principles, however has a different focus of improving the hydrological simulations using parameter regionalization, aligning the models by using consistent river network and topography data, and improving the river channel bathymetry used by the flood inundation model.

Despite the importance of flood risk, many other climate change impact assessments focusing on flooding have not explicitly modeled flood inundation, and instead look at changes in extreme precipitation alone or at changes in river flows. Firstly, it is common to relate future projections of precipitation, simulated by climate models, with flood risk (Frame et al., 2020; Lin

et al., 2018; Cho et al., 2016; Betts et al., 2018). In particular cases, precipitation can be a good proxy for pluvial flooding caused by short-duration, small-scale intense rainfall (Frame et al., 2020). However, more generally, changes in flood events cannot be directly inferred from changes in precipitation alone, as factors such as antecedent conditions (e.g., soil moisture) and channel routing effects also influence the catchment response to climate change (Sharma et al., 2018). Secondly, many studies used hydrological models to quantify water resources and flood risk and assessed how they may change under different climate change projections (Lehner et al., 2006; Dankers and Feyen, 2008; Mohammed et al., 2017). Most of these studies, however, linked simulated runoff or river discharge to flood risk without explicitly simulating flood inundation. This is informative with regards to the chance of a river flooding somewhere, but not for estimating the risk of specific properties flooding, which can only be done using flood inundation models.

Because of these considerations, a global modeling framework that produces a cascade from meteorological information of a precipitation event through to hydrological modeling and high resolution flood inundation modeling steps is very valuable. Here we propose a method for modeling each of the key steps using global, publicly available data-sets in a consistent manner. This allows the production of flood inundation simulations anywhere in the world at high resolution (up to 90 m), even where there are sparse observations of precipitation and river flows. For this study, we construct a model framework for simulating fluvial flooding over the Ganges-Brahmaputra-Meghna (GBM) river basins. This is a large and complex trans-boundary river system, passing through parts of India, China, Nepal, Myanmar and the whole of Bangladesh where it flows into the Bay of Bengal.

Figure 1 shows a schematic of the modeling steps used to translate meteorological information into flood inundation information that is relevant to impacts on different sectors of society. A corresponding example of output from the modeling chain for a flood event in a small catchment is shown in Fig. 2. The modeling steps in this framework are:

a) Meteorological input preparation (section 2)

b) Rainfall-runoff modeling (section 4): We rely on the modular modeling framework FUSE (Framework for Understanding Structural Errors, Clark et al., 2008; Henn et al., 2015) to create an ensemble of conceptual hydrological models,

c) River routing (section 5): We use a stand-alone river routing tool (mizuRoute, Mizukami et al., 2016) to determine flow along river channels,

d) Flood inundation modeling (section 6): We use a 2D flood inundation model (LISFLOOD-FP, Bates et al., 2010) to determine flood hazard.

The interaction between the flood hazard from a particular event with the vulnerability of the infrastructure or populations exposed to that flood determines the impacts which occur. The flood hazard output can be used as a basis for determining impacts to different sectors, which may have different exposure and vulnerability to floods (for example, impacts on human lives, property, industry, agriculture or transport networks). Hence Fig. 1e) gives an indication of possible impacts that may result from the flood hazard modeled here. Due to the diverse nature of these impacts, they are not modeled in this framework, but we highlight the wide applicability of the flood hazard output.

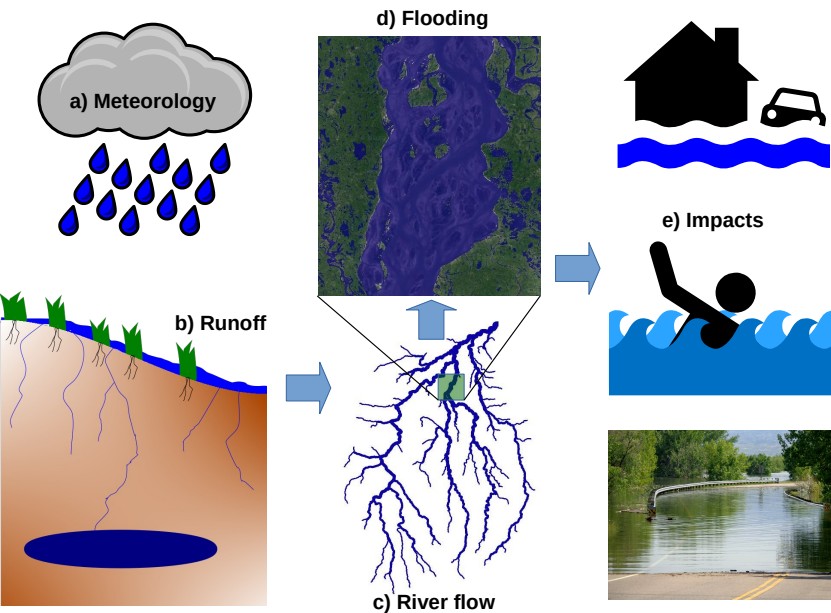

**Figure 1.** Schematic of modeling steps to determine flood impacts from meteorological forcings. The impacts in panel e) (not modeled in this study), can be related to different sectors e.g., human health, transport, agriculture, property damage.

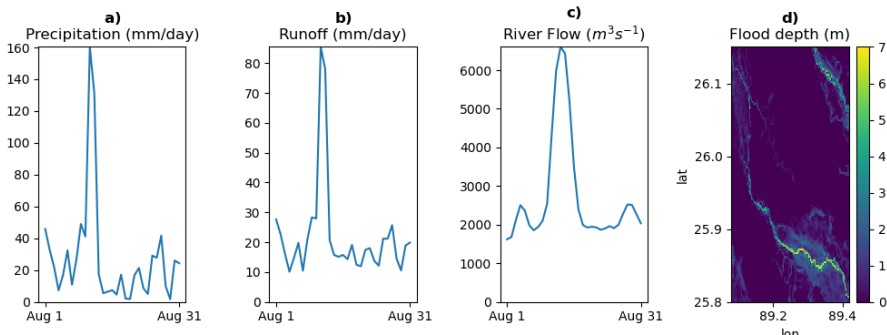

**Figure 2.** Visualization of model data: a) Precipitation (input), b) runoff from FUSE, c) river flow from mizuRoute and d) flood depth from LISFLOOD-FP. Results are for the Teesta river, a tributary of the Brahmaputra river, for a flood event in August 2017. The precipitation and runoff values are catchment averages based on 0.1° resolution gridded data. The discharge shown is the value from the downstream section of this catchment (before joining the Brahmaputra). The flood depth is the maximum over August 2017 calculated at ~270 m resolution (shown over the downstream part of this catchment).

The inputs to these models are critical to the ability to represent realistic flood events. The meteorological inputs are cascaded through the model chain. The models also need to be configured and calibrated to minimize errors introduced at each modeling

step (Pappenberger et al., 2012; Pianosi and Wagener, 2016). The other key part of the modeling framework is producing a realistic river network for use by the river routing and flood inundation models (section 3). The quality of the representation of the river channels and the DEM (describing the floodplain topography) are essential to obtain plausible flood inundation estimates, regardless of the inputs to the models. In this study, we use the MERIT DEM (Yamazaki et al., 2017) which is version of the Shuttle Radar Topography Mission SRTM (Farr et al., 2007) data-set, corrected to remove vegetation and other artifacts. An alternative global DEM with higher resolution and precision is TanDEM-X Krieger et al. (2007), however it includes vegetation surface artifacts, so the corrected MERIT DEM is currently more appropriate for flood inundation modeling (until suitable processed versions of TanDEM-X area available).

We evaluate the performance of the different modeling components used in this study in sections 7–9. Section 7 shows results from the calibration of the FUSE river runoff modeling. Section 8 compares modeled river flow against station based discharge measurements from three downstream gauges on the Ganges, Brahmaputra and Meghna rivers relating to the combined FUSE-mizuRoute simulations. Finally section 9 evaluates LISFLOOD-FP modeled flood inundation extents. Conclusions are presented in section 10.

## 2   Meterological input data-sets

An important factor in producing reliable hydrological and flood inundation simulations is high quality meteorological input data. The FUSE runoff modeling requires precipitation, temperature and potential evapotranspiration (PET). If an unrealistic precipitation data-set is used which misses the peak rainfall intensities, the resulting flood events are likely to be poorly represented. An example of this is the precipitation from a short flash flood lasting one hour will be averaged out in a daily precipitation data-set. The region of the GBM spans a wide area, and a large portion of this has sparse ground observations for precipitation and much of the data is not available publicly. Over the regions where there are gaps in gauge networks, we have less confidence in the gridded precipitation data-sets. Due to the uncertainty in the meteorological observations, we use multiple different input data-sets to establish the variation of model simulations across a range of forcings. We also run simulations using two horizontal resolutions: 0.5° and 0.1°.

The key precipitation data-sets used in this study are from the 'EartH2Observe, WFDEI and ERA-Interim data Merged and Bias-corrected for ISIMIP' data-set (EWEMBI, Frieler et al., 2017; Lange, 2016) and the 'Multi-Source Weighted-Ensemble Precipitation' data-set version 2.2 (MSWEP2-2, Beck et al., 2017, 2019). The MSWEP data-set in particular, aims to improve upon other available precipitation data-sets by merging the best sources of information depending on availability and quality (Beck et al., 2017, 2019). We note that EWEMBI is a multi-variable data-set, while MSWEP is for precipitation only.

The temperature inputs were either taken from EWEMBI or the ERA5 reanalysis (Hersbach et al., 2020). As the ERA5 data-set is provided at 0.25° resolution, it was downscaled to 0.1° and bias-corrected using the WorldClim2 data-set (Fick and Hijmans, 2017) using the method from Hempel et al. (2013). This corrected the monthly mean precipitation and temperature to the WorldClim2 climatology, and corrected the range between the daily mean and minimum/maximum air temperatures. We

did not correct the daily variability as WorldClim2 is a monthly data-set. A summary of the three choices of precipitation and temperature forcing data-sets used in this study is given in table 1.

The calculation of PET is made using the MetSim software package (Bennett et al., 2020). MetSim is based on MtClim and the pre-processor from version 4 of the VIC hydrologic model (Bohn et al., 2013). It requires inputs of minimum and maximum daily temperature in addition to precipitation and calculates PET based on the equation in Priestley and Taylor (1972).

**Table 1.** List of meteorological forcing data-sets used.

| Name | Precipitation | Temperature | Resolution | Time Period |
|------|---------------|-------------|------------|-------------|
| EWEMBI | EWEMBI | EWEMBI | $0.5°$ | 1979–2013 |
| MSWEPp5deg | MSWEP2-2 | EWEMBI | $0.5°$ | 1979–2013 |
| MSWEPp1deg | MSWEP2-2 | ERA5-WorldClim2* | $0.1°$ | 1979–2017 |

*ERA5-WorldClim2 refers to ERA5 bias corrected using the WorldClim2 climatology.

## 3 A consistent river network for simulating streamflow and flood inundation

### 3.1 Generating the river network

Before constructing any part of our hydrological modeling chain, we need to define our network of river channels and catchments. The river network is used for both the river routing and flood inundation modeling (e.g., Fig. 1, panels c and d). Using a river network that is consistent across the modeling components will help improve the reliability of our simulations.

To make this modeling framework widely applicable, we restrict ourselves to global data-sets, and choose the recently developed MERIT-hydro data-set (Yamazaki et al., 2019). MERIT-hydro provides drainage directions and upstream area (referred to as accumulation) at 3 arc-second horizontal resolution (~90 m). MERIT-hydro is largely based on and is designed to be consistent with the MERIT DEM (Yamazaki et al., 2017). To make it appropriate for hydrological modeling, corrections were applied to the digital elevation model such as filling in artificial depressions corresponding to errors in the elevation data.

MERIT-Hydro has been designed to improve on a commonly used global data-set HydroSheds (Lehner and Grill, 2013) by combining multiple global data-sets representing water bodies to accurately determine location of streams. For the flood inundation modeling terrain data we use the MERIT DEM, as ensuring consistency between the DEM and the river network should prevent mismatches between what is being simulated by the models and any artifacts arising from these differences.

To produce the river network, we use the data for the accumulation and flow directions from MERIT-Hydro. These were 145 processed into a vector data-set representing the stream network and catchments using the TauDEM toolbox (http://hydrology.usu.edu/taudem), with additional processing of the data-sets done using LFPtools (Sosa et al., 2020).

When generating the river network, only rivers with an upstream area of greater than 250 km$^2$ were included. This captures larger streams and rivers, however will not include smaller streams. Figure 3a shows an example of this, with smaller streams with a upstream area of 10 km$^2$ also shown. In some situations, including smaller streams can improve the representation of

150 flood events Rajib et al. (2020), however as we are using a coarse resolution DEM over a flat region, there will likely be errors in the representation of some smaller streams. Because of this, we decided to restrict our river network to larger rivers.

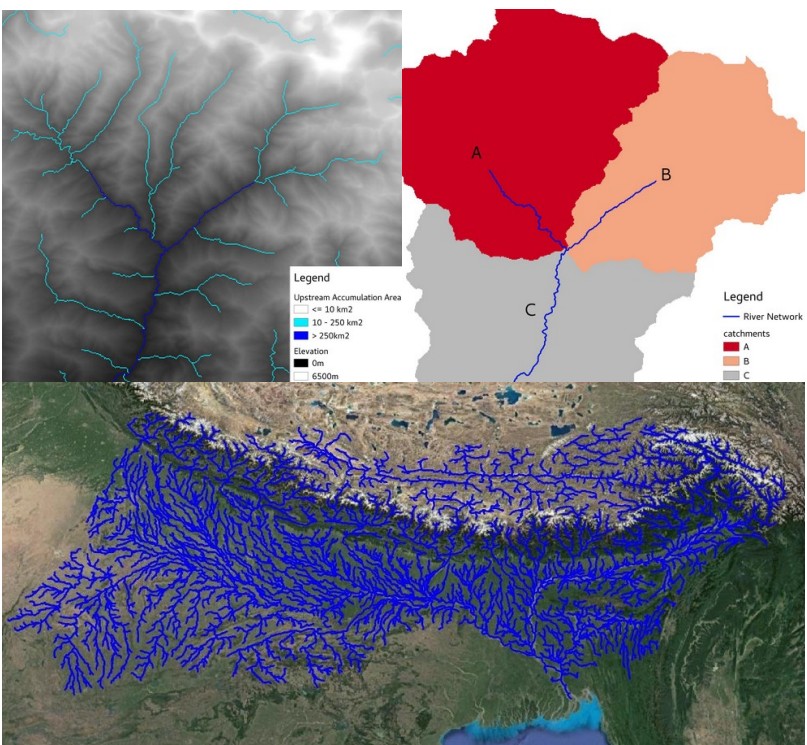

**Figure 3.** MERIT-Hydro GBM Stream Network. a) Example of river points determined from upstream area values. b) Corresponding vector stream network and drainage catchments (using upstream area threshold of 250 km$^2$). c) Map showing the full river network over the GBM basin. Map data ©2020 Google.

The TauDEM toolbox was then used to determine the extent of each river basin in the domain which flows into the ocean and produce a vector network of river segments. Each river segment has a corresponding catchment area which drains into it (Fig. 3b). The catchment areas of each river segment are used by the river routing model to assign a particular amount of runoff
to each segment.

### 3.2 Coarsening river network

The LISFLOOD-FP model is a grid based model. This can be run at the full resolution of the MERIT-hydro data-set (3 arc-seconds, ~90 m), however for large scale applications this may prove to be computationally infeasible. The approach used here is to coarsen the river network and run LISFLOOD-FP at a lower resolution. We still run the mizuRoute model using the
160 original full resolution hydrography as it is discretized at the river segment and catchment scale and not gridded.

Coarsening of the river network needs to be done in a way that does not modify the topology of the river network (i.e. each river tributary should be connected in the same way in the full resolution and coarsened version). For this study, we developed a new network coarsening algorithm, described in Appendix A. This coarsens the input data-sets, then traces down the river network (from low to high accumulation values) and determines which points to include in the coarsened river network and their connections. We note that this process is prone to errors if coarsened too far. For example at lower resolution, multiple streams may flow through the same grid-cell making them indistinguishable. In this study, the choice of coarsening to (9 arc-seconds, ~270 m) was able to significantly reduce the flood model computation, while avoiding the above-mentioned errors.

## 4 Runoff Modeling

Hydrological models are simulation codes representing the terrestrial part of the water cycle and used to transform precipitation into runoff (e.g., Fig 1b). These models include processes such as snow accumulation and melt, infiltration and evaporation in order to represent the storage and transfer of water across the landscape. A multitude of hydrological models exist and rely on different structures (i.e. systems of equations) to represent water dynamics (e.g., Hrachowitz and Clark, 2017). To reflect some of this diversity, we use the Framework for Understanding Structural Errors (FUSE; Clark et al., 2008). FUSE is a framework for conceptual hydrological models, rather than a single hydrological model. It contains building blocks or modules representing different processes used in hydrological models, which can be combined in different ways to produce different model structures. This modularity enables us to generate an ensemble of hydrological models by using a single modeling framework, in contrast to other hydroclimatic impacts studies, which rely on traditional (fixed-structure) hydrological models that have to be setup and run one by one (e.g., Addor et al., 2014; Eisner et al., 2017; Thober et al., 2018). In other words, FUSE is used to sample the structural uncertainty of hydrological models.

### 4.1 FUSE model structures, spatial discretization and inputs

The recombination of FUSE modules leads to over a thousand unique model structures. For this study, we selected three structures (identified by the codes '900','902' and '904'), which mimic three commonly-used hydrological models: HEC-HMS (HEC, 2016), VIC (Liang et al., 1994) and the Sacramento soil moisture accounting model (Burnash et al., 1973), respectively. Each of these structures relies on a different set of equations, whose parameters have to be estimated to reflect landscape features (e.g. soil type and land cover) influencing the hydrological response to meteorological forcing. We use several parameter sets for each model structure (see section 4.2) to account for the uncertainty in these parameter values. This modeling setup hence allows for a quantification of the uncertainty in the hydrological modeling and a separation of the uncertainty due to model processes and model calibration (i.e. structural and parameteric uncertainty). The version of FUSE used in this modeling cascade was set up over the Ganges-Brahmaputra-Meghna river basins. This domain was divided into a regular grid of 0.5° or 0.1° grid-cells and FUSE was run in each grid-cell, treating each grid-cell as a catchment.

Our setup of FUSE requires two types of inputs. It firstly requires gridded meteorological forcing data. The meteorological inputs are precipitation, near-surface air temperature and PET, as per section 2. Secondly, FUSE requires a description of the

terrain within each grid-cell or catchment. To describe the terrain, each grid-cell is split into 16 elevation bands, which enables FUSE to account for the influence of elevation on temperature and precipitation when simulating the snow pack. Based on the full resolution MERIT-DEM data, we calculate the mean elevation and area fraction (of the total grid-cell area) for each elevation band.

## 4.2 FUSE parameter regionalization

Hydrological models require calibration of their parameters to produce realistic runoff for a given catchment (Sorooshian et al., 1993; Perrin et al., 2007). This is commonly performed using observed streamflow (e.g., Falter et al., 2016; Mohammed et al., 2017). For this region, we have access to downstream gauges for the Ganges (Hardinge Bridge station), Brahmaputra (Bahadurabad station), and Meghna (Bhairab Bazar station). FUSE contains a total of 29 parameters which can be calibrated (22 as per Clark et al. (2008) and 7 introduced in Henn et al. (2015)), although not all are in use for each model structure. The GBM is a highly heterogeneous catchment, so the parameters need to be spatially variable. Calibrating all model parameters over such a large region, based on only the downstream gauges which we have access to, would severely under-constrain the parameters. While it may be possible to calibrate the model and obtain accurate discharge at the outlet, the heterogeneity in rainfall-runoff behavior within the catchment would not be realistically represented. In addition to the above considerations for the GBM region, a global model needs to be applicable to ungauged basins (Sivapalan, 2003; Hrachowitz et al., 2013). Hence we chose to use a parameter regionalization scheme using global data to calibrate FUSE. The observed streamflow data for the Ganges, Brahmaputra, and Meghna were only used in validating the results from the combined FUSE-mizuRoute modeled streamflow.

There are many parameter regionalization schemes (Merz and Blöschl, 2004; Samaniego et al., 2010; Parajka et al., 2013) to determine hydrological model parameters in ungauged basins, and here we use the method of Beck et al. (2016). This method involves transferring calibrated parameter sets from small catchments (<10,000 km$^2$) with reliable streamflow records ('donor' catchments) to catchments with similar climatic and physiographic characteristics where the streamflow is not available. The characteristics and data-sets we used are presented in Table 2. These are largely similar to those used in Beck et al. (2016), except that we used newer versions of the data-sets.

The catchments considered for the parameter regionalization were 701 catchments from the Global Runoff Data Centre (GRDC) database with quality controlled, observed discharge. These catchments are a subset of those used in Beck et al. (2016) (We did not have access to the Falcone et al. (2010) GAGES-II data-set or the Peel et al. (2000) data-set for this study and there were eight GRDC catchments which we did not process because of issues with meta-data). Note that there are no donor catchments within the GBM region which is simulated for this study. Three donor catchments were matched to each grid-cell to include a measure of parameter uncertainty. This resulted in 73 donor catchments for the 0.5° grid and 222 donor catchments for the 0.1° grid. In both cases, we only considered donor catchments which were used for at least 50 grid-cells.

We note that there are differences in scale between the catchments and the model grid-cells. The catchments used have areas ranging between 17 km$^2$ and 10,000 km$^2$, with an median area of 1,300 km$^2$. On the other hand, the grid-cells have areas of roughly 120 km$^2$ and 3,000 km$^2$ at the equator, for the 0.1° and 0.5° resolution grids, respectively. Catchments with similar

**Table 2.** List of catchment characteristics used for parameter regionalization, and corresponding data sources.

| Characteristic | Source | Data Resolution |
|---|---|---|
| Precipitation | WorldClim2 (Fick and Hijmans, 2017) | 30s (~1 km$^2$) |
| Temperature | WorldClim2 | 30s (~1 km$^2$) |
| PET | WorldClim2, (calculated using Hargreaves and Samani, 1985) | 30s (~1 km$^2$) |
| Aridity index | Worldclim2 (calculated as precipitation/PET) | 30s (~1 km$^2$) |
| Forest cover | Global Forest Change, (Hansen et al., 2013) | 30s (~1 km$^2$) |
| Snow cover | MODIS MOD10CM v6 (Hall and Riggs, 2015) | 0.05° (~5 km) |
| Terrain Slope | MERIT DEM ((Yamazaki et al., 2017, , slope calculated from elevation)) | 3s (~90 m) |
| Soil clay fraction | Soilgrids250m (Hengl et al., 2017) | 250 m |

size to the grid-cells are most appropriate for this type of regionalization, however, we considered all available catchments with area less than 10,000 km$^2$ to ensure our sample of catchments covers the widest possible range of climatic and physiographic characteristics.

For each catchment, FUSE was set up as a single lumped model where forcing inputs were averaged over the catchment areas. FUSE was then calibrated using the shuffled complex evolution algorithm (Duan et al., 1992) to minimize the RMSE of modeled streamflow against observed. The time period run was determined by the overlap between the meteorological observation timeseries and the streamflow timeseries, and half of the data was used for calibration. For regionalization purposes, the donor catchments were simulated using the same meteorological forcing data-sets as the gridded model and different

calibrations were performed for each of the FUSE model structures used here.

## 5   River routing modeling

The river routing model used in this framework (e.g., Fig 1c) is mizuRoute (Mizukami et al., 2016). mizuRoute is a stand-alone one-dimensional (1D) river routing model. It is configured to take runoff from a hydrological model as input. Then it applies two stages of routing. The first is hillslope routing, which represents the delay from runoff being produced at any location in

the catchment and it reaching the river channel. The second is 1D river channel routing where water is routed downstream through the river network. mizuRoute has two options of physics schemes for the channel routing. The scheme used here is the impulse response function - unit-hydrograph scheme.

### 5.1   Model configuration

mizuRoute relies on three main input data-sets to compute the river flow for each river segment. These are the river network,

the runoff data, and a mapping between the hydrological model grid and the catchments used in mizuRoute.

Firstly it takes in the river network topology. As mizuRoute uses a 1D routing algorithm, rather than needing the full spatial information of the river network, it is discretized at the river segment and catchment level, defined per section 3. For each river segment, the model needs various information representing its physical properties. For example, mizuRoute requires the slope and length of each segment, information about the downstream segments and the area of each drainage catchment.

## 5.2 Mapping runoff to river catchments

mizuRoute is designed in a way which enables it to read the runoff output from a gridded hydrological model such as FUSE. So in this model set-up, the FUSE output for each simulation can be input directly into mizuRoute without further processing.

To handle the difference in grids between the rectangular grid used in FUSE, and the irregularly shaped drainage catchments used in mizuRoute, we need one additional data-set which produces a mapping between each catchment, and the FUSE grid-cells which overlap it. Where multiple grid-cells overlap a catchment, an area weighted sum is used to determine the runoff for that catchment. Figure 4 shows an example of the overlap between a particular catchment and the FUSE model grid-cells. As this mapping is a fundamental property of the model grid/network, it only needs to be calculated once for a particular model configuration.

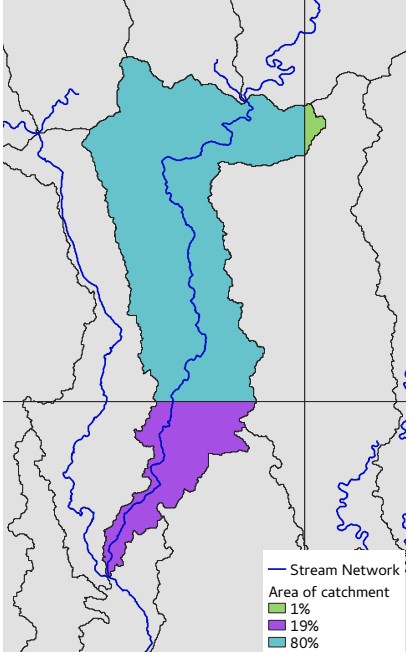

**Figure 4.** Example of the grid-to-catchment mapping procedure. A single drainage catchment within the GBM basin is highlighted here. This particular catchment overlaps with three FUSE model grid-cells, so runoff from these three cells is used in this catchment. The runoff is weighted by the fractional area of each cell in the catchment (percentage areas shown in legend).

## 6 Flood inundation modeling

The flood hazard model used for this study (e.g., Fig 1d) is LISFLOOD-FP, a 1D-2D flood inundation model (Bates et al., 2010; Neal et al., 2012). LISFLOOD-FP solves the shallow water equations, without advection, at each location on the flood plain to explicitly simulate the floodwater flow. We use a sub-grid version of LISFLOOD-FP, where river channel flow is computed in 1D and floodplain flow in 2D is calculated for water when it exceeds the river bank height within the 1D channels.

Flood simulations were run using discharge inputs from mizuRoute for the period April–October each year. This covers the 265 monsoon season over which the major floods in Bangladesh occur.

### 6.1 New LISFLOOD-FP features for this study

For this study, we have made some updates to the LISFLOOD-FP model to improve its applicability for large scale flood inundation modeling using the MERIT-Hydro hydrography.

In previous versions of LISFLOOD-FP (Neal et al., 2012), the model had been restricted to allowing river channels to flow 270 in only the four cardinal directions (referred to as d4). However, river networks such as used in Hydrosheds and MERIT-Hydro specify channels flowing in the diagonal directions as well (referred to as d8). For this study, we developed a version of LISFLOOD-FP that uses d8 directions for river channel flow. This allowed a more accurate representation of the MERIT-Hydro river network. In previous studies e.g., Sosa et al. (2020), the river network was first converted to d4, which can be prone to errors. The errors in converting from d8 to d4 directions are also greater at coarser resolutions, so using d8 directions makes 275 the procedure of coarsening the river network more reliable.

In this LISFLOOD-FP version, we also allow the specification of a new input data-set for the sub-grid river channels. This is a data-set specifying links between cells in the river network. By default, LISFLOOD-FP computes flow between any neighboring sub-grid channel cells. When using this option, only specified cells are linked together. This is particularly relevant for coarser versions of the model, where different tributaries may become close together, or in meandering river sections. Also, 280 when using d8 directions, there can be many possible links between sub-grid cells which are not necessarily desirable, making this a useful inclusion to the model.

### 6.2 Input Data-sets

This model requires a representation of both the physical properties of the river channel and the flood plain. For the flood plain flow, the main data-set is the Digital Elevation Model (in this case MERIT-DEM).

For the river channels, additional information is required to represent the sub-grid properties of the channel. The model uses the following data-sets to represent the river channel bathymetry:

– Elevation of River Bank (determined from MERIT-Hydro elevation)

– Width of River Channel (determined from remotely sensed data-sets)

– Elevation of River Bed (See below)

### 6.2.1 Estimating river bank elevations from the DEM

The LISFLOOD-FP model requires elevation values for the river bank at each point along the sub-grid channel. Importantly, the bank elevation will not necessarily be the same as the DEM elevation, particularly where the river width is wider than the grid size. In, these cases the DEM elevation may represent a particular water elevation rather than the bank. To help mitigate this problem, when determining river bank elevations, we use a river channel mask from the MERIT-hydro data-set. For each point in our river network, we select the closest point in the DEM outside the channel mask to represent the bank elevation.

### 6.2.2 Identifying river widths

For river widths we use information from remote sensing based data-sets. In this study, we used the Global River Widths from Landsat (GRWL, Allen and Pavelsky, 2018) data-set. This data-set does not locate the center points of river channels in the same location as the MERIT-Hydro data-set, so widths are determined by locating the closest point in the GRWL data-set to each river channel (within a certain search radius). Where river widths cannot be identified, they are interpolated between points along the river, or assumed to be a small river and assigned a width of 30 m when that is not possible.

We also note that the MERIT-Hydro data-set provides a river width variable. This would be simple to include in this set-up, as the river widths are specified at the same location as the MERIT-Hydro based river channels that are used for this study. However, for braided rivers such as used in this case-study, MERIT-Hydro only captures the width from individual braids, whereas it is more appropriate to use the combined width across all the braid channels.

### 6.2.3 Calculation of river depths, bank and bed elevation

To determine the capacity of the sub-grid river channel, it is important to have a reasonable estimate of the river depths at each point. The river widths are determined from a remote sensing product, however data for river channel depths is not readily available. One common and simple method for estimating river depths is by using the assumption that a river water level reaches the bank height at a particular 'bankfull' frequency (e.g., once every two years is a classical geomorphologic approximation). River depths that are consistent with the flow at this bankfull frequency can then be approximated.

One way of approximating the depth is using the Manning's equation e.g., used in Sampson et al. (2015). This requires river flow, width and slope. The width and slope are already available, and using the assumption that the river water level reaches the bank we can estimate a bankfull river flow. The bankfull flow used here was chosen as the two year return period flow in mizuRoute simulations for the 1980-2013 period. Discharge values at each node in the river network were linearly interpolated to each river grid point.

Another method used in this framework to compute the bankfull water elevations and river depths that are more consistent with the bank heights is using a gradually varying flow solver (Neal et al., 2021). This solves for the water elevation along whole reaches and accounts for backwater effects, whereas the Manning's equation approach solves each channel point independently and assumes uniform flow. The solver has been shown to produce more realistic bathymetry and hence flood depth. It reduces over-prediction of flood extent which is possible using the Manning's method due to the neglect of backwater effects.

For this study, we use the gradually varying flow solver to find the set of river bed elevations that give water elevations at 1 in 2 year return period discharge that best match the bank elevations extracted from the DEM. Depths calculated using the Manning's equation were used as first guesses of the bathymetry. In the LISFLOOD-FP model, the calculated bed elevations and bankfull water surface elevation were used for the bed and bank elevations. The initial selection of points from the DEM may not represent the actual river bank elevation, but other locations in or near the channel, or contain errors from vegetation artifacts or noise in the DEM. The choice of using the computed water surface corresponding to the bankfull flow instead of the bank heights chosen from the DEM is intended to produce smoother bank heights and reduce the influence of errors in the DEM.

## 6.3 Boundary conditions

The main boundary conditions for LISFLOOD-FP are determined by the river network and the river flow calculated by the mizuRoute model.

### 6.3.1 Boundary conditions at edge of domain

At the domain boundary, various boundary conditions are possible in LISFLOOD-FP. For this study, we set the downstream boundaries to be determined by the slope of the river segments. Here the free water surface slope between the final river cell and the boundary is set to a constant (which is the average slope along the river segment). This is an approximation, as the water surface slope changes over time, and will vary during a flood event. However, without any more reliable local information about each river segment, this approximation gives a reasonable estimate based on the slope of the terrain. A buffer region on either side of each downstream river point is given the same boundary condition to allow water to flow out of the domain from the flood plain.

All other regions of the boundary (most importantly the upstream boundary points) are set to be closed. This ensures that any inflows go into the model domain.

### 6.3.2 Inflows

Daily inflows from the mizuRoute model are inserted at specific points of the LISFLOOD-FP domain in river channel grid-cells. As mizuRoute flows for each river segment represent the outflow at the downstream end of that segment, some assumptions were made to allow for inserting the river flow at the upstream edges of the domain. mizuRoute provides two inputs that are used here: 1. 'streamflow': streamflow calculated, including flow from higher sections of the river network, and 2. 'routed runoff': runoff from a specific drainage catchment, routed into the corresponding river segment. We note that for headwater catchments (i.e. with no upstream tributaries), the streamflow and routed runoff are equivalent.

As the LISFLOOD-FP model routes the river flow, we only want to use the mizuRoute streamflow for the upstream boundaries of the LISFLOOD-FP model domain. At all other locations within the domain, the output of the mizuRoute routed runoff is used.

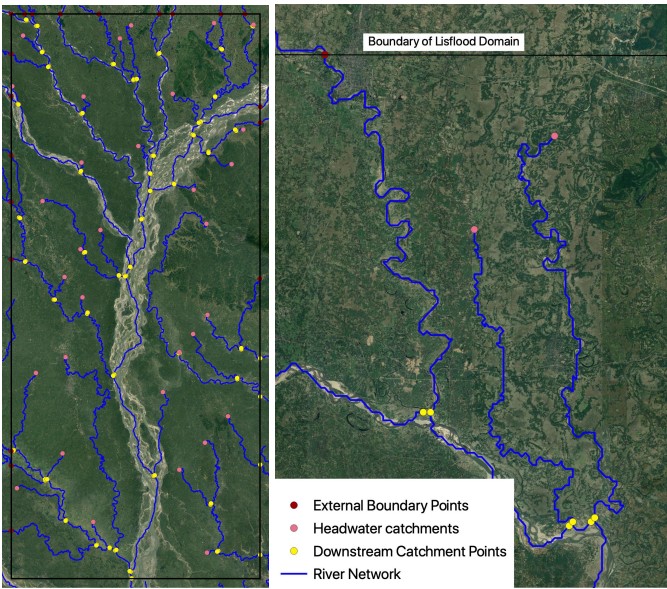

**Figure 5.** Left: Example of stream network and LISFLOOD-FP input points for subdomain of the GBM: 89.08-90.3E,24-26.5N. Right: zoomed in section of this region, showing part of the northern boundary. Map data ©2020 Google.

The following possible cases are possible for river flow inputs.

1. Headwater river segment of tributary is within the LISFLOOD-FP domain,

2. Headwater river segment enters at the edge of the LISFLOOD-FP domain,

3. Non-headwater river segment continues outside the LISFLOOD-FP domain,

4. Non-headwater river segment is entirely within the LISFLOOD-FP domain.

Each of these cases is handled slightly differently with regards to how the mizuRoute discharge is input into the LISFLOOD-FP model domain.

For headwater catchments, discharge is input at two points: at the upstream point and the downstream point of the stream. The routed runoff from mizuRoute is split up between these two, based on the upstream accumulation area of each of these two points:

$$D_u = R * A_u / A_d \tag{1}$$

$$D_d = R * (A_d - A_u) / A_d \tag{2}$$

Where $D_u$ and $D_d$ are upstream and downstream input discharge. $R$ is the mizuRoute routed runoff, and $A_u$ and $A_d$ are the upstream area for the upstream and downstream points respectively ($A_d$ is the catchment area). We note that these equations

are also used when the upstream end of a headwater catchment is outside the domain. In this case the $D_u$ and $A_u$ are values for the point at the edge of the domain.

For non-headwater streams which continue upstream out of our LISFLOOD-FP domain, we have an boundary condition at the edge of the domain which is determined from the mizuRoute streamflow. For these catchments, we also insert a contribution from the routed runoff at the downstream point of the reach (weighted by the portion of the catchment inside the domain).

$$D_e = S - R * (A_d - A_e)/(A_d - A_u) \tag{3}$$
$$D_d = R * (A_d - A_e)/(A_d - A_u) \tag{4}$$

Here $S$ is the mizuRoute streamflow for this catchment. $D_e$ is the the input discharge at the edge of the domain. $A_u$ is the upstream area of the upstream point of the tributary (outside the LISFLOOD-FP domain), so $A_d - A_u$ is the full catchment area, and $A_e$ is the upstream area of the tributary at the edge of the LISFLOOD-FP domain.

For (non-headwater) streams which are entirely within the domain, discharge is only input at the downstream end. For these catchments the discharge added is simply:

$$D_d = R \tag{5}$$

## 7 Evaluation of FUSE parameter regionalization

The evaluation of the regionalization and FUSE-mizuRoute part of the model chain is performed in two steps because of the lack of streamflow observations in the GBM river basin. First, we focus on the parameter regionalization, presenting two main factors which determine its performance: how closely the characteristics of donor catchments match those of the grid-cells simulated in the GBM region and how well each of the donor catchments are calibrated. The analysis in this section utilizes streamflow observations from the donor catchments (none of which are in the GBM region). We secondly use streamflow observations from the GBM region to evaluate the skill of the FUSE-mizuRoute simulations and the benefits of the regionalization over uncalibrated simulations in Section 8.

Fig. 6 shows a normalized measure of dissimilarity (as per Beck et al., 2016), between the catchment characteristics used for the regionalization in section 4.2. Here a value close to zero indicates that a donor catchment has been found with very similar characteristics to those of the grid-cell. We note that the regionalization has successfully found similar donor catchments over much over this domain with the exception of a few regions. Notably the Tibetan plateau and some of the Himalayas are not easily matched to donor catchments. Part of the Meghalaya mountains are where the worst matches are found. This region is very steep and mountainous as well as having extremely high monsoon precipitation.

Another thing to note about the dissimilarity measure here, is that it is based on the characteristics in table 2. However, grid-cells with zero dissimilarity may not necessarily be ideally matched to their donor catchment(s) due to differences in other catchment characteristics not considered here. Conversely, a high dissimilarity may be a reflection of the choice of characteristics rather than a reflection of how applicable donor catchment parameters are. For example, Beck et al. (2020) chose

to use square-root transformed precipitation as a more normally distributed characteristic. This may reduce the dissimilarity calculated for regions such as the Meghalaya mountains while still selecting suitable donor catchments.

To evaluate the calibration of donor catchments used in the regionalization method, we use the Nash-Sutcliffe efficiency score (NSE) as a metric. In this metric, a calibration that perfectly matches the observed flow would give a value of 1. Fig. 7 shows the NSE scores from the calibration simulations of donor catchments (showing the average of NSE scores for the best three donor catchments transferred to each GBM grid-cell). This value differs for each FUSE model structure, as the structures will give different performance depending on the particular catchment. We note that this regionalized NSE score is

an artificial measure and expect the performance of individual grid-cells in the GBM region to be lower than the calibration scores in Fig. 7. This is because the calibration scores of the donor catchments also account for local random errors in forcing and local idiosyncrasies in rainfall-runoff behavior – two factors that regionalization cannot account for. With comparison to Fig. 6, in Fig. 7 the Tibetan plateau is again problematic, but some mountainous regions which show low similarity show good performance in the donor catchments which were matched to.

In this study, we did not use the skill scores in the selection of donor catchments. This meant that the catchments with closest characteristics were used and the same donor catchments were used for all simulations at the same resolution. However, this has the drawback that poorly calibrated parameter sets may be used. Choosing only high performing donor catchments would be ideal, but does have the drawback of reducing the sample size which may increase the dissimilarity between donors and grid-cells. In addition, calibration scores of donor catchments vary across forcing data-sets and FUSE structures, so this would

result in choosing different donor catchments for each configuration. This would complicate the set-up, making it harder to determine the reason for differences in performance between configurations.

In the absence of streamflow observations for small catchments within the GBM, we are not able to evaluate the FUSE model performance at the grid-cell level. Instead, Figs 6 and 7 give a qualitative indication of where the regionalization scheme may find reasonable parameters sets or have deficiencies. However, this region is highly heterogeneous and it not possible to

predict from the regionalization how skillful the simulation of the whole region will be. Thus, our main evaluation of FUSE simulations considers the FUSE-mizuRoute routed flow for the large GBM catchments where we have observed streamflow.

## 8    Evaluation of routed streamflow in the Ganges-Brahmaputra-Meghna

We evaluate the streamflow produced by the FUSE-mizuRoute model output against downstream flow data for the Ganges, Brahmaputra and Meghna rivers. These streamflow measurements are not used in the model calibration so provide an indepen-

dent verification of the skill of our model set-up. The closest node in the mizuRoute network to each station was determined for this comparison.

Fig. 8 shows a comparison of the FUSE-mizuRoute simulated river discharge against station observations over the 1980-2013 period. Left hand plots show the average climatology of daily discharge between April and November, covering the rainy season. Three observationally based precipitation data-sets are used to drive the models in these plots as in section 2. For the

climatology, the seasonal cycle is captured by simulations using all of the precipitation data-sets. In the Ganges, the MSWEP

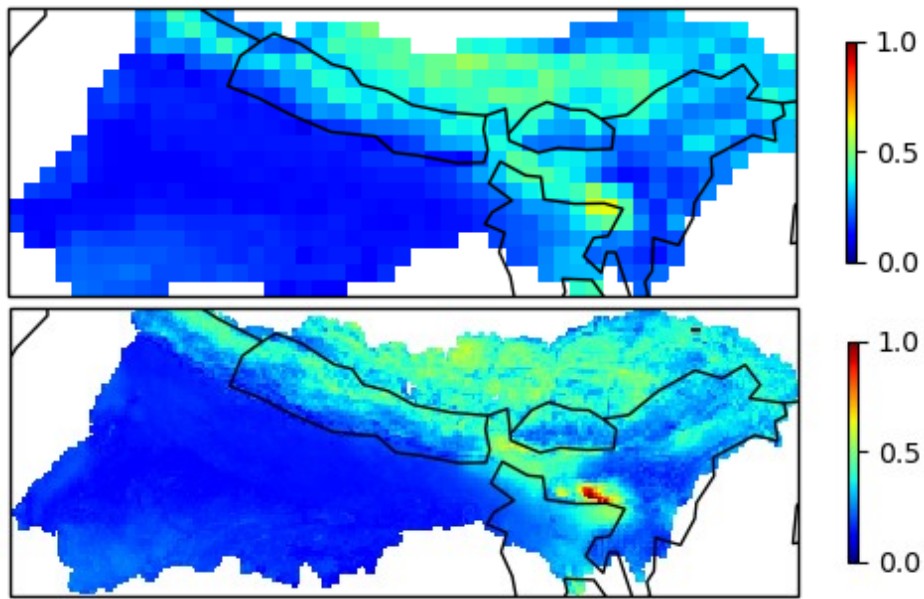

**Figure 6.** Average of dissimilarity measures between grid-cells in the GBM basin and the 10 best matching donor catchments. A value of 0 indicates a perfect match between the catchment characteristics of the donor and the grid-cell. Top is 0.5° and bottom is 0.1° horizontal resolution.

data-sets seem to over-predict the discharge, and in the Meghna river, the peak in the seasonal cycle occurs early relative to the observations.

The right hand panels in Fig. 8 shows the yearly maximum discharge for different return periods. For the Brahmaputra and Ganges, these show that the maximum flows correspond well with the observed discharge, especially given the uncertainties in these values.

For the Meghna river, despite the mean flow being of the right magnitude, the peak flows are completely unrealistic. We note that wetlands (called Haors) take up a large portion of the upstream portion of the Meghna river basin. So rather than the runoff flowing through a simple 1D river channel as mizuRoute simulates, much of the water stays in the wetlands, attenuating the flow. This is a common problem with river routing models, requiring an approach to represent floodplain storage or otherwise parametrise wetland processes (Zhao et al., 2017; Dadson et al., 2010; Fleischmann et al., 2018). An approach to test this would be to run LISFLOOD-FP over the whole Meghna river basin. As LISFLOOD-FP computes water inundation and flow in two dimensions, it should be able to capture this effect of flood water transfer between river channels and wetlands. However due to computational restraints, a LISFLOOD-FP simulation over the whole catchment hasn't been done for this study. It should also be possible to implement a wetland module as part of the mizuRoute setup e.g., Rahman et al. (2016). Another possible approach that may improve mizuRoute performance in this situation would be to additionally represent channel bifurcation (e.g., Ikeuchi et al., 2015). Implementation of these schemes would requires a different approach to this study, or additional local knowledge for calibration and manual specification of the wetland areas. Because of this limita-

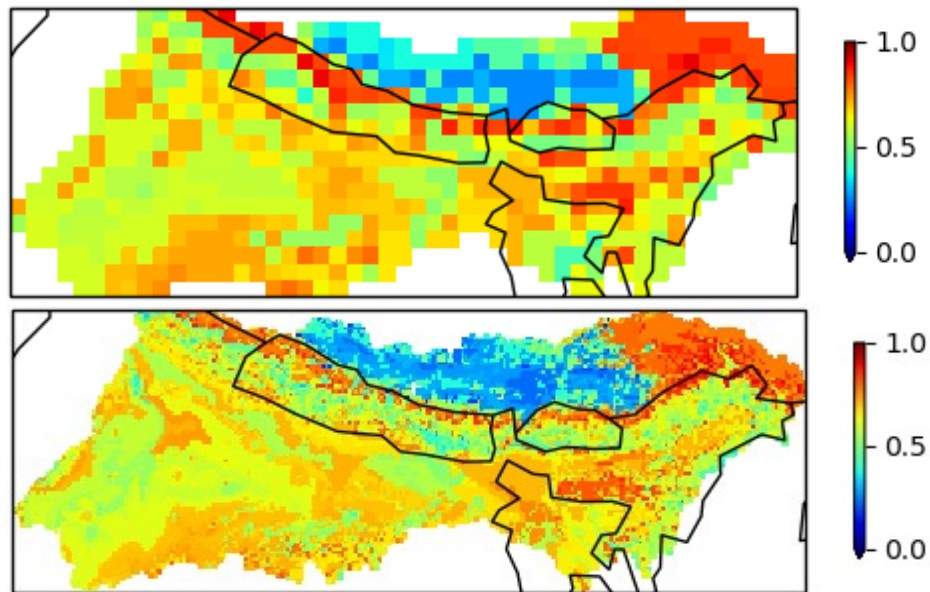

**Figure 7.** Mean of NSE scores from 3 best donor catchments transferred by regionalization to each grid-cell. The NSE values were calculated using observed streamflow from each donor catchment (all outside the GBM) hence should not be interpreted as the model performance in each grid-cell of the GBM region. Top is 0.5° and bottom is 0.1° horizontal resolution. NOTE: plots are for simulations using FUSE structure 902, forced by MSWEP precipitation. A value of 1 indicates perfect agreement between the model and observations (in the donor catchments), while a value of zero indicates that the model has no more skill than using the mean discharge.

tion of the FUSE-mizuRoute model simulating the Meghna river, we focus on the results from the other two basins (Ganges and Brahmaputra) in the following evaluation.

Table 3 shows Kling-Gupta efficiency (KGE) scores for uncalibrated simulations and the minimum, median and maximum KGE scores across a range of parameter-sets for discharge on the Ganges and Brahmaputra rivers. We note that the FUSE structure 900 is the best performing FUSE structure on the Brahmaputra river, but the best performing FUSE stucture differs between forcing data-sets on the Ganges. The best performing simulations have a Kling-Gupta efficiency score of over 0.8 for both catchments, with higher scores seen over the Brahmaputra river. In most cases, the worst performing calibrated simulations

give improvements upon the uncalibrated simulations, but there are a few simulations e.g., for FUSE structure 902 over the Ganges where the uncalibrated simulations give similar KGE scores to the median of the calibrated simulations.

In addition to the climatology, Fig. 9 show a comparison of streamflow for two specific years, 1988 and 1993. Here we compare observations against simulations using default parameters and calibrated simulations as well as using inputs from EWEMBI or MSWEPp1deg data-sets. These show the improvement gained by using the parameter regionalization, but also

highlight that the uncertainty in the meteorological forcing data is of the same or greater magnitude than the calibration uncertainty in the runoff model.

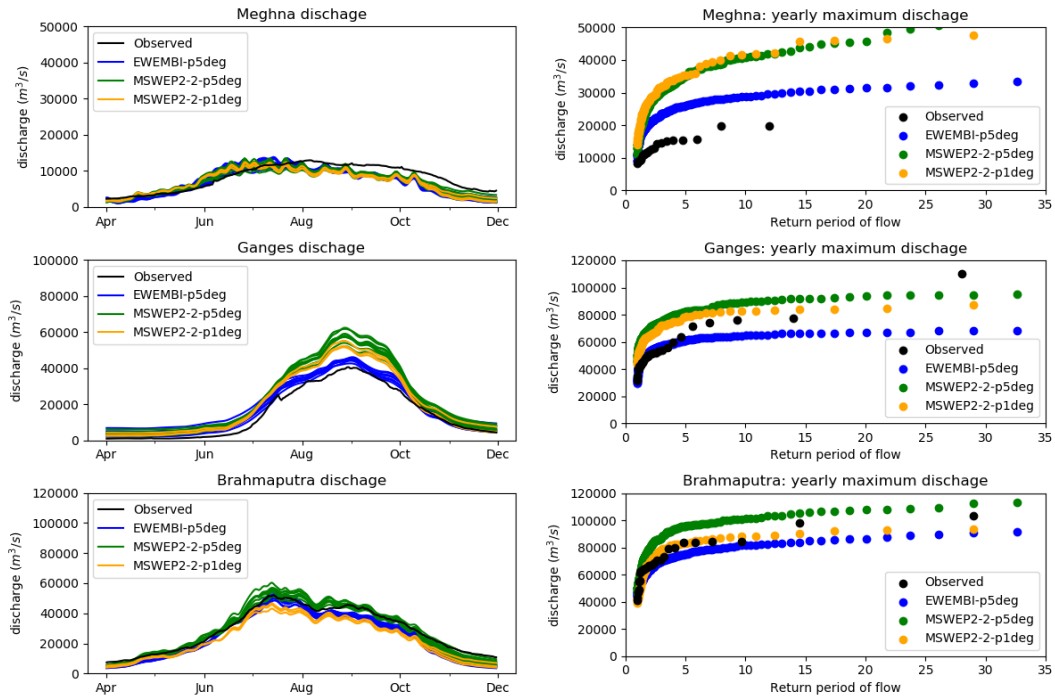

**Figure 8.** Comparison between observed and modeled streamflow for three stations on the Meghna (top), Ganges (middle) and Brahmaputra (bottom). Left plots show seasonal cycle and right plots show return periods for yearly maximum discharge. Different lines in the same colour show different FUSE structures with different parameter sets.

## 9 Evaluation of flood inundation in the Ganges-Brahmaputra-Meghna

In the climate community, gridded observation data are available at a range of time frequencies. However, this is not currently the case for flood inundation data. Evaluation of simulated flood inundation is often a complicated procedure. Accurately

mapping flood extents using ground observations over large areas would be an extremely labor intensive and impractical effort. Instead, satellite images are often employed to estimate flood extents, and can do so at the high spatial resolution needed to verify flood model inundation. There are, however, drawbacks in using satellite observations and these differ by the satellite product used.

For an estimate of historical flood extents, we firstly use a Global Surface Water data-set (GSW, Pekel et al., 2016), produced

by the European Commission's Joint Research Centre. This has been produced by combining around three million Landsat images. From this data-set, we use a recurrence product, which gives the percentage of years that each grid-cell was inundated (at ~30 m resolution) between 1984 and 2015. We note that flooding is obscured by clouds in the Landsat images, and also may not be detected under dense vegetation. Furthermore, this data-set has sparse temporal coverage, with one valid data-point per month or less over the majority of our modeled region. Because of the sparse sampling frequency of this data-set, we expect

it to provide a low estimate of the actual flooded extent. Short duration floods, particularly in small, flashy catchments will

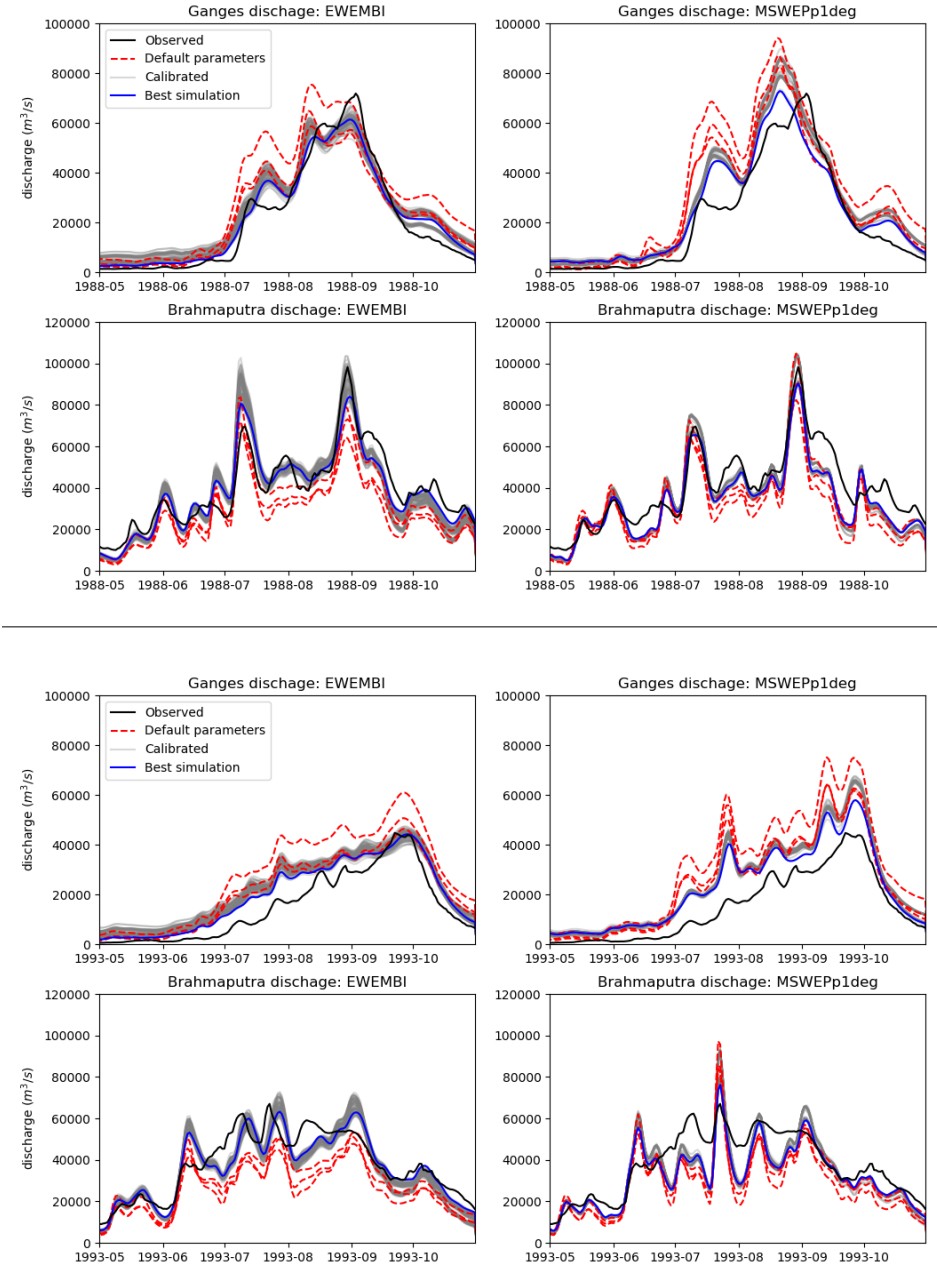

**Figure 9.** Yearly streamflow for 1988 (top) and 1993 (bottom). Each of these plots has four panels with EWEMBI simulated streamflow on the left and MSWEPp1deg simulated streamflow on the right. Top panels show streamflow for the Ganges and bottom panels show streamflow for the Brahmaputra river. In each of the panels the observed station discharge is compared to different simulated flows. Red dashed lines show simulations using default FUSE parameters. Grey lines show simulations using a range of different calibrated parameters. Blue lines show the simulation with the highest Kling-Gupta efficiency score calculated over the 1980-2013 period ('Best simulation').

**Table 3.** Kling-Gupta Efficiency (KGE) scores for mizuRoute simulations, categorized by input precipitation data-set and FUSE structure. KGE scores are given for simulations with default parameters (uncalibrated), then minimum, median and maximum KGE scores from simulations with different calibrated parameter sets. Note, KGE scores are calculated over the rainy season (April-Oct) between 1985-2013.

| Forcing data-set | FUSE structure | Uncalibrated KGE | Min KGE | Median KGE | Max KGE |
|---|---|---|---|---|---|
| **Ganges** | | | | | |
| EWEMBI | 900 | 0.40 | 0.53 | 0.61 | 0.67 |
| EWEMBI | 902 | 0.73 | 0.70 | 0.75 | 0.81 |
| EWEMBI | 904 | 0.71 | 0.69 | 0.74 | 0.78 |
| MSWEPp5deg | 900 | 0.15 | 0.34 | 0.44 | 0.53 |
| MSWEPp5deg | 902 | 0.50 | 0.42 | 0.49 | 0.60 |
| MSWEPp5deg | 904 | 0.51 | 0.48 | 0.54 | 0.60 |
| MSWEPp1deg | 900 | 0.24 | 0.58 | 0.58 | 0.59 |
| MSWEPp1deg | 902 | 0.59 | 0.58 | 0.59 | 0.62 |
| MSWEPp1deg | 904 | 0.56 | 0.67 | 0.67 | 0.68 |
| **Brahmaputra** | | | | | |
| EWEMBI | 900 | 0.67 | 0.81 | 0.84 | 0.86 |
| EWEMBI | 902 | 0.68 | 0.80 | 0.83 | 0.85 |
| EWEMBI | 904 | 0.56 | 0.73 | 0.75 | 0.77 |
| MSWEPp5deg | 900 | 0.75 | 0.81 | 0.86 | 0.88 |
| MSWEPp5deg | 902 | 0.72 | 0.75 | 0.80 | 0.83 |
| MSWEPp5deg | 904 | 0.53 | 0.70 | 0.77 | 0.81 |
| MSWEPp1deg | 900 | 0.72 | 0.75 | 0.75 | 0.77 |
| MSWEPp1deg | 902 | 0.68 | 0.70 | 0.71 | 0.73 |
| MSWEPp1deg | 904 | 0.50 | 0.66 | 0.67 | 0.69 |

often occur and recede in between observations and not be detected. Because of this, we use the GSW recurrence data-set to estimate commonly flooded locations, which will be relevant to the major rivers, but do not consider it an authoritative record of all flooding that has occurred.

We secondly verify the flood model inundation extent against Sentinel-1 synthetic aperture radar (SAR) images (referred to as 'Sentinel'). SAR can be used to detect water surfaces as they are often smoother than the surrounding terrain. Compared to optical images, SAR has the advantage that it can see through clouds, however it also has limitations. For example, flooded areas underneath vegetation or with emergent vegetation (e.g. rice paddies) or that is roughened by strong winds can be incorrectly detected as dry. Conversely, smooth wet vegetation (e.g. flattened wet grass) can be incorrectly detected as water. SAR estimates of flood area can correctly classify up to around around 75% of the true flood extent (Horritt et al., 2001). For

this comparison, we estimate the flooded area by a simple thresholding approach, where the SAR amplitude is less than -16dB (threshold chosen as per Uhe et al. (2019)).

The flood model was run and evaluated over the region shown in Fig. 5. This is a section of the Brahmaputra river along with a number of tributaries. This includes the entirety of the Brahmaputra river (Jamuna) within Bangladesh, until 20km before its confluence with the Ganges (Padma) river. Furthermore, this region provides a good example of a flood captured during 2017

in the Sentinel-1 observations. It is complicated braided section of river which is challenging to capture accurately in a model. We note that in this domain, the rivers here are generally flowing north-to-south. This is a relatively small domain, however it is computationally feasible to run this modeling framework over a larger region, for example the whole of Bangladesh.

Fig. 10 shows the flood recurrence for both the GSW data-set and the flood inundation model. The model produces additional flooding in the upstream reaches of the domain compared to the GSW data-set. This is not surprising, given the limitations in the

495 GSW data-set discussed above. We also note that from year-to-year, the braids of the river channels shift due to sedimentation and moving sand-banks, which is captured in the GSW data-set. The flood model, however uses a DEM which is static, and largely based on NASA's SRTM data-set from 2000, so will not capture the year-to-year variation in the river channels. Another consideration of this set-up is that the sub-grid river channels in the flood model only flow though a single 'main' stem. Flooding of the braids is a result of the 2D floodplain flow which only occurs after the water level in the sub-grid channel

reaches bank-height. That said, the model flood extent over the Brahmaputra river's braids shows a good agreement with the GSW data-set.

To take a more quantitative comparison between the flood model and GSW, we take a specific flood recurrence of 20%, corresponding to a 1-in-5 year flood event. Given the short period of 32 years of observations, more extreme events will be less certain, so this event was chosen as a significant event, but relatively well constrained. Fig. 11(a) shows the same data as

in Fig. 10, but comparing the data-sets for the 1 in 5 year flood inundation extent.

To highlight the influence of differing resolutions, we additionally aggregate the GSW data-set to the same (~270 m) resolution as the flood model by considering the 270 m grid-cells flooded if at least 10% of the corresponding high resolution cells are flooded. This is shown in (Fig. 11b). As the GSW data-set captures flooding at 10 times finer scales than the flood model, the flooding of a small river with area less than a model grid-cell will appear to show over-estimation of flooding by the model

no matter how good the simulation, when compared at 30 m. This has the effect of reducing the area of model flood where GSW is not flooded, but also increases the area of GSW flooding where the model is dry in Fig 11b).

Another limitations of the above comparison is that, as previously mentioned, short duration floods may be missed in the GSW data-set. This is a particular consideration in the north western tributaries of this region, which are flashy catchments fed by rainfall off the Himalayan foothills. The GSW data-set will also detect standing water such as wetlands, rice-paddys or

515 resulting from pluvial flooding rather than fluvial flooding, so some regions (particularly the smaller scale flooding) where the model is under-predicting the GSW flood may be due to this.

We additionally analyze a major flood event in 2017, starting around 11th August. The Brahmaputra river reached a record water level during this event, so it was a very extreme event. We use two Sentinel images for this event, one from 12th August (early in the event) and one on the 24th of August (late in the event), shown in Fig. 12. The SAR data has a resolution of 10 m,

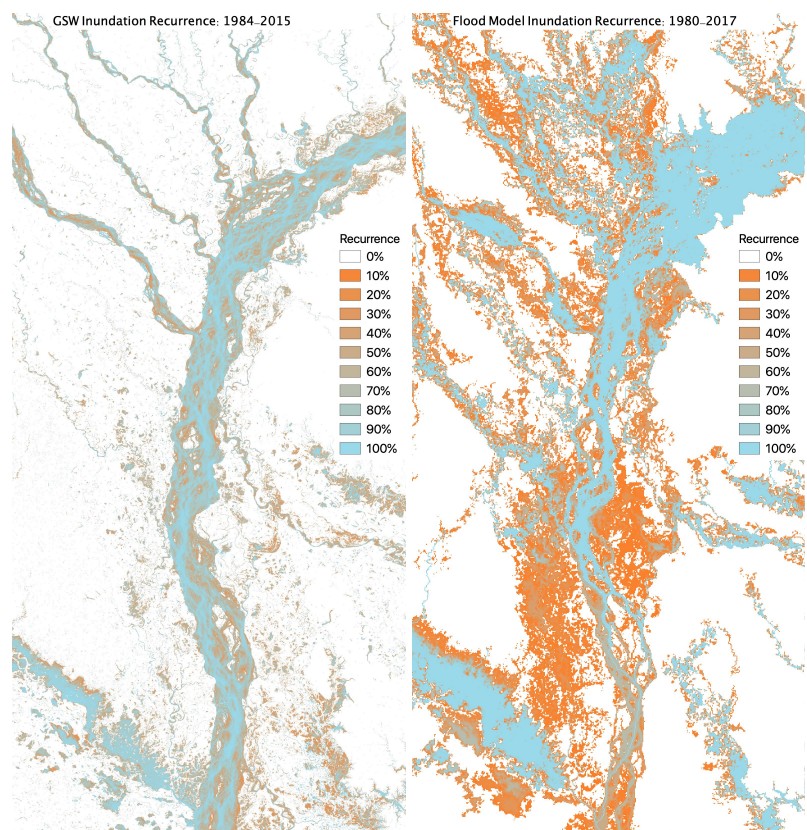

**Figure 10.** Recurrence (percentage of years) of maximum flood inundation for the GSW data-set (left) and the flood model forced by MSWEPp1deg precipitation. Model results show flood inundation greater than 0.5m, over the period 1980-2017, with 9 simulations run for each year, generated from runoff simulations with differing FUSE configurations.

but there is significant noise at the grid-cell level, so the Sentinel data was aggregated to 270 m as per the GSW data. In the left panel we see that a much larger proportion of the region is inundated, with better agreement between the model and Sentinel in the upstream tributaries. This image may also indicate that the flood model may be under-predicting the actual flood extent, although we note that some of the flooding detected by Sentinel may be a result wet vegetation rather than true flooding. Other flooding not captured by the model may be due to flooding of small streams that are not resolved by our model or pluvial flooding caused by a combination of high rainfall and poor drainage over the flat floodplains. The second panel shows a picture after much of the floodwaters have receded, although there is some additional flooding compared to August 12 along the main stem of the Brahmaputra. This picture is much more consistent with the GSW data-set, again indicating the limitation of the GSW data-set, in that the sampling of flood events has missed the peak floods.

To summarize, while the flood model is not matching all of the observed flood extent, many features of the observed flooding are captured. Disagreement between the model and satellite observations have a number of causes. Locations of flooded rice-paddys, wetlands, flooding of small streams and pluvial flooding may not be captured in the model. Sentinel may have false

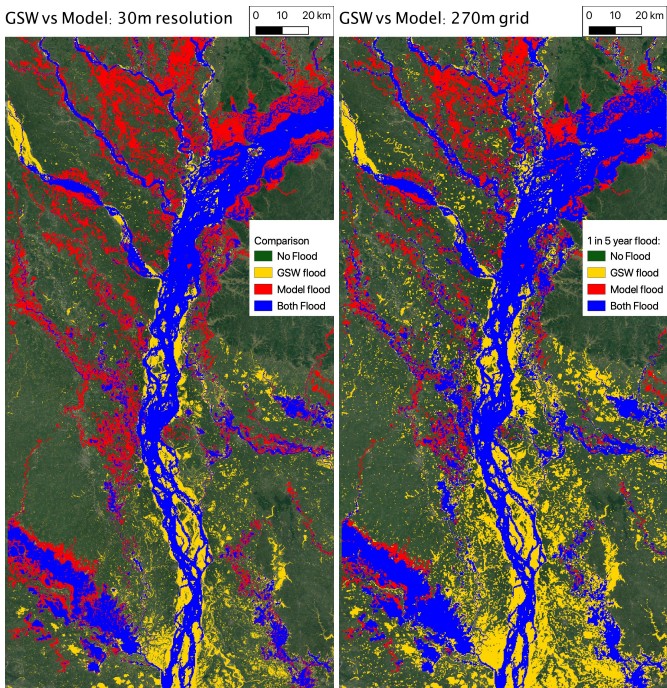

**Figure 11.** Flood extent comparisons between model and GSW data-set. a) Flood extent of a 1 in 5 year (20% recurrence), comparing modeled and GSW flood extent at 30 m resolution. b) As per (a), but at 270 m resolution. In (b), the 270 m grid-cells for GSW are considered flooded where 10% of higher resolution cells are flooded. Note: Model results show flood inundation greater than 0.5m. Modeled flood recurrence is calculated over the period 1980-2017, with 9 simulations run for each year, generated from runoff simulations with differing FUSE configurations. Map data ©2020 Google.

positive and negative detections of flood relating to vegetation, and the GSW data-set is likely to miss a large amount of flooding in small catchments.

The uncertainties in the model outputs mean that when using this model cascade, for example to assess climate change impacts, it would not be appropriate to focus on changes for individual pixels. Rather, changes aggregated to river reach or sub-catchment will be more robust.

## 10   Conclusions

We introduce here a new model framework which allows simulation of high resolution flood inundation based on meteorological inputs that could come from either observations or climate models. Results driven by three observationally based meteorological data-sets are evaluated, showing good agreement with streamflow for the Brahmaputra and Ganges rivers, as well as producing realistic flood inundation over a section of the Brahmaputra river and its tributaries.

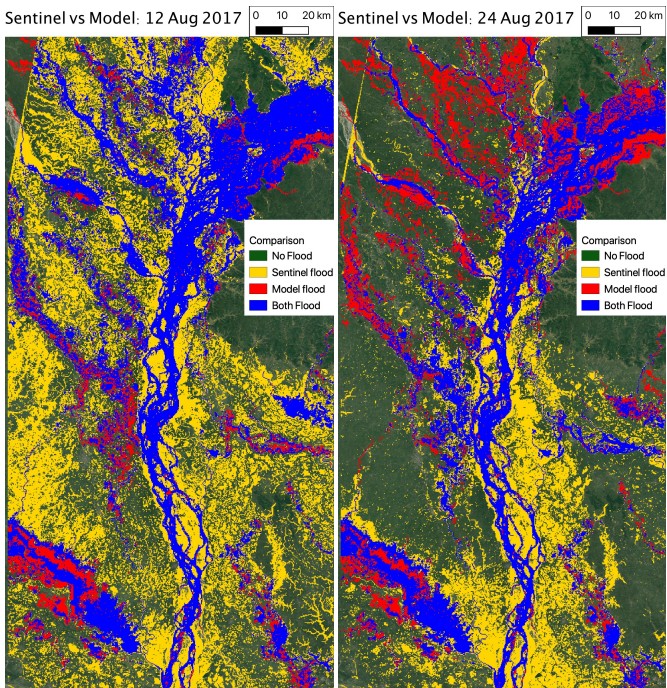

**Figure 12.** Flood extent comparisons between model and Sentinel observations for August 2017 flood event. a) Sentinel image for August 12, 2017. b) Sentinel image for August 24, 2017. Both panels are compared against the maximum extent of flood inundation greater than 0.5 m in a single simulation for 2017. The 270 m grid-cells for Sentinel are considered flooded where 10 % of the higher resolution cells are flooded. Map data ©2020 Google.

As a summary of the model steps, Table 4 describes the inputs and outputs that cascade through the modeling components. Each of these steps also rely on a number of assumptions. For example, FUSE is a lumped hydrological model which parametrizes the varied hydrological processes which occur within each grid-cell. It does not take data-sets containing soil or vegetation information as input, instead relies on calibration of parameters to account for these variations. mizuRoute is a simple 1D routing model, which is efficient for running over large scales, but does not take backwater effects or floodplain storage into account. In theory the FUSE runoff could instead be input directly into the LISFLOOD-FP river channels to include more realistic river channel dynamics, however running the LISFLOOD-FP model over the whole GBM basin was computationally infeasible for this application. In the LISFLOOD-FP model, there are also approximations required to represent the river channel bathymetry. The river widths are estimated from remote sensing data, and bank elevations are estimated from the DEM. However in absence of reliable data, river depths are estimated using the geomorphologic assumption of bankfull flow once every two years in combination with a gradually varied flow solver, and river bed friction coefficients are also assumed to be constant. All these assumptions contribute towards uncertainties in the modeling framework, however the uncertainties in external inputs such as precipitation and DEM mean that introducing extra complexity in these models will not necessarily produce better results.

**Table 4.** Modeling inputs and outputs (at daily frequency*), which are passed between models.

| Step | Inputs | Outputs |
|---|---|---|
| Meteorology | N/A | Precipitation, temperature (mean, minimum, maximum) |
| MetSIM | precipitation, temperature (minimum,maximum) | potential evapotranspiration |
| FUSE | precipitation, temperature (mean), potential evapotranspiration | runoff |
| mizuRoute | runoff | hillslope routed runoff (by catchment), streamflow (routed from all upstream reaches) |
| LISFLOOD-FP | hillslope routed runoff (added to rivers internally), streamflow (at domain boundaries only) | water depth, sub-grid river channel and floodplain water flows |

*LISFLOOD-FP model uses a dynamic time-step (~10 seconds here), allowing higher frequency output if required.

The flood inundation model is the most compute-intensive component of this modeling chain. So the ability to compute flood inundation at high resolution over relatively small domains, while computing streamflow for a whole river basin to produce realistic upstream boundary conditions, provides a efficient and flexible way of determining local flood risks. Of course, with expected increases in future compute power, there is also potential to compute flood inundation risks over whole basins.

There are disadvantages in using a complex modeling chain, as each model introduces its own set of uncertainties, making the factors influencing the accuracy of the final flood inundation hard to interpret. As part of this study, we look into the uncertainty at different steps. We run our modeling framework using different observation-based meteorological data-sets, and also consider three different methods of representing the hydrological processes using FUSE. This gives a range of different plausible simulations to compare. Another possible disadvantage in using a modeling chain with a standalone hydrological

model for climate change projections is that the hydrological model does not feed back to the climate. For example, in a fully coupled earth system model, the modeled evapotranspiration influences the climate simulation which will not occur in our framework. The model set up has also been tested in a challenging location for simulating flood inundation. The wetlands in the Meghna basin were problematic for the FUSE-mizuRoute setup, which was not able to capture a realistic peak discharge for that region. Future work in similar environments will benefit from considering the processes necessary to simulate river

flow through wetlands. These finding highlight the importance of validating global models across diverse regions and only making conclusions where we have high confidence in the model performance.

We additionally note that the use of global data-sets in this framework is an advantage in terms of applicability. As new global DEMs, precipitation data-sets or regionalization methods are developed, they will drive improvements in the hydrological and flood modeling simulations. However, models which have access to high quality local data have the potential to outperform a

purely global model. Thus this method using parameter regionalization represents a middle ground between flood model chains calibrated using local information (e.g., Falter et al., 2016; Grimaldi et al., 2019; Rajib et al., 2020), and global flood models forced by uncalibrated hydrological models (e.g., Winsemius et al., 2013; Dottori et al., 2016). The regions where this model

framework will give the most value is where there are gaps in the availability of input data-sets such as meteorological data, river gauge observations or high quality LIDAR based DEMs. This represents the poorest regions of the world, which may be highly vulnerable and need the greatest help in adapting to climate change.

River (or fluvial) flooding which is captured in this modeling framework is just one type of flooding. A comprehensive flood risk approach would also assess the other types of flood hazard. These types of flooding relate to different types of meteorological or hydrological events and require different modeling approaches. Table 5 summarizes these types of flooding. Operational flood forecasting centers are required to produce forecasts for all of these types of flooding. However due to the differences in scales and data required, climate change studies into flood risk generally only consider one of these types of flood risk, with very few studies explicitly simulating compound flooding such as combined coastal and river flooding (Zscheischler et al., 2018; Bevacqua et al., 2019; Pasquier et al., 2019; Bates et al., 2021). This methodology has the potential for being used for climate change research or for flood forecasting (for regions where high quality hydrological data-sets do not exists). Extending this framework to include other types of flooding will be a useful extension for future work. Pluvial flooding in particular should be a relatively simple addition to this framework although pluvial flooding occurs at smaller scales, this would require both higher resolution DEMs and precipitation than fluvial flooding. Including pluvial flooding may also need to be done in coordination with simulating fluvial flooding in smaller watercourses.

**Table 5.** Types of flood hazards and key modeling inputs.

| Type | Description | Modeling requirements* |
|------|-------------|------------------------|
| Fluvial | Flooding caused by river over-topping banks and inundating floodplain | Representation of river channels and streamflow |
| Pluvial | Flash flooding caused by short very intense precipitation events. This may include flooding of streams and small rivers not resolved by the fluvial model | High resolution (temporal and spatial) precipitation |
| Coastal | Storm surge or tidal inundation of coastal areas | Coastal boundary conditions |
| Groundwater | Flooding caused by groundwater/water table rising | Representation of groundwater and interactions |

*High resolution modeling of all flood types require high quality digital elevation models to describe the terrain and floodplain.

This paper presents a methodology for global modeling of high resolution flood inundation. The use of flood risk information at scales less than 1 km has the potential to transform how climate change risks relating to flooding are presented. Instead of using general measures of precipitation or river flow, flood inundation can be used to inform about specific risks. These risks can relate to people or properties exposed to floods, area of farmland inundated or locations where roads or railways are impacted by floodwaters. Future work will include driving this model cascade with climate change projections to investigate future flood hazard. These global flood models can then give better information about projected damage from climate change, to communities rich and poor.

*Code availability.* The flood-cascade source code is archived on Zenodo (Uhe, 2020). The LISFLOOD-FP code archived on Zenodo (LISFLOOD-FP developers, 2020) for non-commercial use only.

## Appendix A: Algorithm for Coarsening River Network

The method here uses the upstream area (acc) and the Strahler order (ord, Strahler, 1957) of each river section. The Strahler order is an integer which is one for headwater streams, when going downstream it is incremented when two streams with the same order join. The Strahler order is calculated by TauDEM when generating the stream network.

The main steps for the coarsening algorithm are as follows:

1. combine a block of grid-cells together (e.g., 5x5)

2. Require a minimum number of the original cells (e.g., 3) to be part of the river network, otherwise exclude from coarsened network

3. Take maximum values for acc and ord over each block

4. Traces down the river network from upstream to downstream, to determine the next downstream point. For each point, consider all points with the same or greater ord and greater acc as possible downstream points

    (a) In the case where there is no change in tributary or a smaller tributary merges into the current stream, the downstream point will have the smallest acc of these points

    (b) In the case where the current stream merges into a larger stream, we check if there is a large jump between possible downstream points. If there is a large jump in acc (>acc0) between possible downstream points then we assume the smaller value is above where the tributary joins, and use the larger acc value for the downstream point.

This algorithm has been tested coarsening the MERIT-hydro 3 arc-second resolution data-set to 9,15 and 30 arc-seconds. There are limitations to this coarsening process. The biggest one is if two different tributaries approach closely then diverge, they may be incorrectly merged. This occurred in a few locations when coarsening to 30 arc-second resolution in the GBM with the MERIT-hydro hydrography. So the 30 arc-second resolution version would not be appropriate without modification of the hydrography to prevent this occurring. Where two tributaries join a main river close together, the coarsened river networks may also join them at the same grid cell. In the GBM, there were 2 and 10 instances of this occurring in the 9 and 15 arc-second river networks respectively. Other minor changes to the rivers as a result of the coarsening process is that it does in some places straighten out meandering segments which almost turn back on themselves.

*Author contributions.* PU, DM and PB conceived the study. PU conducted modeling and analysis. NA assisted with FUSE modeling. JN assisted with LISFLOOD-FP modeling. HB assisted with meteorological forcing data and parameter regionalizaiton. All authors contributed to interpreting analysis and writing of the manuscript.

*Competing interests.* The authors declare no competing interests

*Acknowledgements.* We acknowledge the Global Runoff Data Centre, 56068 Koblenz, Germany for use of discharge observations. DM was supported by a NERC independent research fellowship (NE/N014057/1). NA acknowledges funding from the Swiss National Science Foundation (fellowship P400P2_180791). PB was supported by a Royal Society Wolfson Research Merit award.

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
