# Peer review of "Model cascade from meteorological drivers to river flood hazard: flood-cascade v1.0"

_Geoscientific Model Development, 2020_

## Referee Comment (RC1) · Anonymous Referee #1 · 15 Jan 2021

General Comments

Firstly I would like to congratulate the authors on producing such a well written manuscript. As someone who knows something about all the components in the model cascade but wouldn't call myself an expert in any of them, I was able to follow the methodology easily. There is a logical progression through each step and the links between models (data transfer, domain/resolution changes, required assumptions etc.) are explicitly explained. I didn't event spot a typo until line 545!

As the authors mention, there are a growing number of attempts in the literature to create model cascades from meteorology to flood inundation. Many of these attempts have remained largely conceptual due to the challenges of identifying consistent data to run each component in the model cascade for global applications. The authors

should therefore be commended for their thoughtful application of the model cascade to the Brahmaputra river and helpful commentary on how they have addressed these issues of data scarcity using available global data sets. The complete case study adds significant value to this manuscript.

Excepting a few minor edits, I would recommend this manuscript for publication.

Specific comments

The justification of this approach is to improve on climate change assessments that use precipitation or river flow as a proxy to flood hazard. The authors should make it clear throughout the abstract and introduction that the model cascade they have designed could be used for this application (or indeed has been designed for this application) but this step isn't included in the current manuscript.

Line 19 I would argue the impact of floods is largely driven by the vulnerability of the population and infrastructure in the locations that they hit rather than the catchment characteristics (see the literature on "no natural disasters"). Climate change assessments (sometimes) take this into account by including projections of increased population etc. This model cascade does not go as far as to model impacts in this sense (ending at flood inundation/hazard) and I think this should be explicitly acknowledged in the text which does stray into discussing impacts in several places. You should check your terminology throughout the manuscript. Also can you justify in the text why you haven't included impact modelling into the cascade?

Whilst I think the following suggestion would increase the impact of the manuscript, I do not consider it an essential pre-requisite to publication as the model cascade is already well described and documented. As a model description paper to enable others to set up a similar approach (and for general appreciation of the significant undertaking this type of model cascade is to set up) I would like to see as assessment/table (which could be descriptive) of how important each step is in the model chain; how much time does it take, is the required data easily available, what assumptions have been made

to make the step possible, and does including it as an explicit modelled step in the cascade notably improve the end result/reduce uncertainty. Although I note a future paper using climate projections is planned, perhaps you could consider a comparison of your results against just using the precipitation or river flow as a proxy within this future work to demonstrate the benefit of using a full model cascade.

Technical corrections

Figure 1 schematic. The last impacts picture isn't very clear, I think it's a collapsed bridge, but better images are probably available. However this manuscript only takes the model cascade as far as inundation modelling, there is no attempt to model vulnerability / impact beyond water depth so I would question if the impact images are misleading here.

~Line 55 I would move the statement from the bottom of page 5 "to investigate changes in future flood risk. . ." to page 2. The introduction could lead the reader to think you are going to present a climate change simulation, I would be explicit early on that you are presenting a model cascade that could support this type of assessment in the future. Similarly I think you could add a line to the end of the abstract along the lines of "this approach could be used to assess the impacts of climate change by. . ... [changing the meteorological drivers]"

Line 145. Could you include evidence of this in the appendix?

Line 199 I suggest "we did not have ACCESS TO the. . ."

Figure 5 ang 6 I suggest adding the lines you already have in the text "a value of 0 indicates. . ." to the figure caption so figures can be interpreted easily.

Line 415 channellings of modelling wetlands / very flat floodplains is a general issue for hydrological models and not unique to this application. Could you cite some other examples to demonstrate the challenge.

Line 465 please specify why this area of the catchment was selected to demonstrate

the inundation modelling

Line 509 what would these limitations mean if this model cascade was being used to assess climate change impacts?

Line 547. The terminology used in the conclusion is slightly confusing / inconsistent i.e. "high resolution flood inundation", "fine scale flood risk information". These terms are relative, what is presented in this manuscript could be described as high resolution compared to what is currently available for flood modelling globally at these type of scales, but it is not high resolution in the more general context. The discussion within the manuscript has previously focussed on the challenges of adequately describing the topography, river network etc in the absence of high resolution data.

Line 545 missing or incorrect word (the = this?)?

Line ~550 would this model cascade be of potential use for flood forecasting in data/resource poor regions of the world or are the run times prohibitive?

Table 4 The definition of pluvial flooding refers to "flash flooding caused by short, very intense precipitation events". Could you clarify if you consider this to be before water enters a watercourse or if you would also include flooding from small watercourses. I think there is potential for confusion in the definition here as the methodology only includes watercourses with catchments >250m2. Including pluvial flooding would require a continuum of approaches to go from catchments of 250m2 to pluvial flooding which many not be as simple as implied in line 545.

---

## Referee Comment (RC2) · Anonymous Referee #2 · 31 Jan 2021

As a person who has been working with all the three components featured in this paper, i.e., meteorology, hydrologic modeling, and flood inundation mapping, I am seriously confused about this flood-cascade. I thoroughly read the paper. Overall, it is very well-written. The methodology is reasonable. Yet, I think that the authors' justification of creating a flood-cascade sends a wrong message to the new-generation of readers, disavowing the long history of hydrologic-hydraulic modeling research (apologies for the use of strong words here).

After reading the abstract and introduction, a reader might think that this "flood-cascade" is a brand-new concept. But that would be wrong. In fact, the so-called flood-cascade, integrating meteorology drivers to flood hazard predictions and hence assessment of climate change impacts, has existed in our scientific community for a

long time with different forms, scales, and names. For example, the GLOFRIM for integrated hydrological–hydrodynamic global modelling by Hoch et al. (2017) and the National Water Model in the United States are just two examples of many existing, globally applicable flood-cascade frameworks. Each of these existing frameworks are well-resolved flood-cascades with cascading input-output structures according to the authors' definition. I hereby strongly oppose the author's narrative in the existing version of the paper.

In summary, the merit of this paper comes down to evaluation of a relatively new LIS-FLOOD modeling framework for one of the world's data-poor flood-prone basins. The paper has all the potential to getting accepted for publication, however, with major changes in the title, abstract, and introduction.

---

## Author Comment (AC2) · 27 Feb 2021

**Reviewer comment**

*As a person who has been working with all the three components featured in this paper, i.e., meteorology, hydrologic modeling, and flood inundation mapping, I am seriously confused about this flood-cascade. I thoroughly read the paper. Overall, it is very well-written. The methodology is reasonable. Yet, I think that the authors' justification of creating a flood-cascade sends a wrong message to the new-generation of readers, disavowing the long history of hydrologic-hydraulic modeling research (apologies for the use of strong words here).*

*After reading the abstract and introduction, a reader might think that this "flood- cas-*

[Figure]

*cade" is a brand-new concept. But that would be wrong. In fact, the so-called flood-cascade, integrating meteorology drivers to flood hazard predictions and hence assessment of climate change impacts, has existed in our scientific community for a long time with different forms, scales, and names. For example, the GLOFRIM for integrated hydrological–hydrodynamic global modelling by Hoch et al. (2017) and the National Water Model in the United States are just two examples of many existing, globally applicable flood-cascade frameworks. Each of these existing frameworks are well-resolved flood-cascades with cascading input-output structures according to the authors' definition. I hereby strongly oppose the author's narrative in the existing version of the paper.*

*In summary, the merit of this paper comes down to evaluation of a relatively new LIS-FLOOD modeling framework for one of the world's data-poor flood-prone basins. The paper has all the potential to getting accepted for publication, however, with major changes in the title, abstract, and introduction.*

**Response**

Thank you for your frank review of our manuscript. We take the major point of this review, that the concept of a model cascade for flood inundation is not new, and there are previous examples of this. It was not our intention to imply otherwise, so we will take this onboard and modify the article to emphasise previous research in this area.

The novelty of this model framework relates to the way the different modelling components have been linked together and the models built and calibrated using globally available data. The techniques used here include improvements compared to previous flood modelling frameworks for data-scarce regions, so this should not simply be dismissed as solely a model evaluation paper.

---

## Author Response (AR1)

Firstly I would like to congratulate the authors on producing such a well written manuscript. As someone who knows something about all the components in the model cascade but wouldn't call myself an expert in any of them, I was able to follow the methodology easily. There is a logical progression through each step and the links between models (data transfer, domain/resolution changes, required assumptions etc.) are explicitly explained. I didn't event spot a typo until line

As the authors mention, there are a growing number of attempts in the literature to create model cascades from meteorology to flood inundation. Many of these attempts have remained largely conceptual due to the challenges of identifying consistent data to run each component in the model cascade for global applications. The authors should therefore be commended for their thoughtful application of the model cascade to the Brahmaputra river and helpful commentary on how they have addressed these issues of data scarcity using available global data sets. The complete case study adds significant value to this manuscript.

**Excepting a few minor edits, I would recommend this manuscript for publication.**

Thank you for taking the time to go through our manuscript in detail and for your very positive review. We will respond to your specific comments below:

**Specific comments**

The justification of this approach is to improve on climate change assessments that use precipitation or river flow as a proxy to flood hazard. The authors should make it clear throughout the abstract and introduction that the model cascade they have designed could be used for this application (or indeed has been designed for this application) but this step isn't included in the current manuscript.

Thank you, we have made this clearer in the revised manuscript. In the abstract we include the following text:

> *This framework is designed to be driven by meteorology from observational data-sets or climate model output, and to be an accessible tool using freely available data. In this study, only observations are used to drive the models, so climate changes are not assessed. However, by comparing current and future simulated climates, this framework can also be used to assess impacts of climate change.*

In addition, we have added the following text into the second paragraph of the introduction:
> *The hydrological cycle is being altered due to the influences of climate change. So making flood inundation models more accessible for use in climate change impacts studies is an important step for future research. The approach presented here, is designed to enable more robust flood inundation estimates using climate model outputs, over larger regions. This study focuses on describing the model framework and evaluating its performance against past observations of river flow and flood inundation. We do not include climate projections here, which will be covered in a future study.*

Line 19: I would argue the impact of floods is largely driven by the vulnerability of the population and infrastructure in the locations that they hit rather than the catchment characteristics (see the literature on "no natural disasters"). Climate change assessments (sometimes) take this into account by including projections of increased population etc. This model cascade does not go as far as to model impacts in this sense (ending at flood inundation/hazard) and I think this should be explicitly acknowledged in the text which does stray into discussing impacts in several places. You should check your terminology throughout the manuscript.

Yes, this sentence should be referring to flood hazard rather than impacts. We have corrected and expanded on this statement (below). We have also checked the correct use of hazard, impacts and risk throughout the manuscript.

Corrected text

*It is these non-linear interactions which determine the ultimate magnitude of the resulting flood hazard (Sharma et al., 2018, Grimaldi et al., 2019). The flood impacts are furthermore the result of the exposure and vulnerability of populations.*

Also can you justify in the text why you haven't included impact modelling into the cascade?

Yes. Our idea with producing the schematic in Fig. 1 was to show that from the hazards that we calculate, there are many possible impacts – e.g., to infrastructure, transport, agriculture, peoples lives/health. Because of the different ways of quantifying these impacts, we would argue that presenting the hazards, and making them available for use in other impacts models can increase the applicability of the results from this approach, rather than quantifying a single type of impact in this framework.

Added new paragraph (line 84):
*The interaction between the flood hazard from a particular event with the vulnerability of the infrastructure or populations exposed to that flood determines the impacts which occur. The flood hazard output can be used as a basis for determining impacts to different sectors, which may have different exposure and vulnerability to floods (for example, impacts on human lives, property, industry, agriculture or transport networks). Hence Fig. 1e gives an indication of possible impacts that may result from the flood hazard modeled here. Due to the diverse nature of these impacts, they are not modeled in this framework, but we highlight the wide applicability of the flood hazard output.*

**Whilst I think the following suggestion would increase the impact of the manuscript, I do not consider it an essential pre-requisite to publication as the model cascade is already well described and documented:**

As a model description paper to enable others to set up a similar approach (and for general appreciation of the significant undertaking this type of model cascade is to set up) I would like to see as assessment/table (which could be descriptive) of how important each step is in the model chain; how much time does it take, is the required data easily available, what assumptions have been made to make the step possible, and does including it as an explicit modelled step in the cascade notably improve the end result/reduce uncertainty.

Thank you for this suggestion. We have produced a table in the conclusions listing the inputs and outputs of each step of the model cascade. We also include a paragraph in the conclusions discussing assumptions relating to different modelling steps.

Although I note a future paper using climate projections is planned, perhaps you could consider a comparison of your results against just using the precipitation or river flow as a proxy within this future work to demonstrate the benefit of using a full model cascade.

Yes, comparing changes in flood hazard from explicit simulations of climate projections with changes from scaling present day flood simulations with projected changes in river discharge, is something we would like to do with this framework.

**Technical corrections**

Figure 1 schematic. The last impacts picture isn't very clear, I think it's a collapsed bridge, but better images are probably available. However this manuscript only takes the model cascade as far as inundation modelling, there is no attempt to model vulnerability / impact beyond water depth so I would question if the impact images are misleading here.

Thank you, that is a good point. We think including the impacts in this schematic are important for motivation of the modelling, but as discussed above, we have made it clearer that the impacts were not modelled for this study. We have swapped the bridge image.

~Line 55 I would move the statement from the bottom of page 5 "to investigate changes in future flood risk. . ." to page 2. The introduction could lead the reader to think you are going to present a climate change simulation, I would be explicit early on that you are presenting a model cascade that could support this type of assessment in the future.
We have moved this statement to the introduction as suggested.

Similarly I think you could add a line to the end of the abstract along the lines of "this approach could be used to assess the impacts of climate change by. . ... [changing the meteorological drivers]"
As above, we added text addressing this.

Line 145. Could you include evidence of this in the appendix?
We have added some text in the appendix, describing the types of errors occurring when coarsening the river network to different resolutions.

Line 199 I suggest "we did not have ACCESS TO the. . ."
We have made this change.

Figure 5 ang 6 I suggest adding the lines you already have in the text "a value of 0 indicates. . ." to the figure caption so figures can be interpreted easily.
Thank you, we have expanded these captions to explain the data values.

Line 415 channellings of modelling wetlands / very flat floodplains is a general issue for hydrological models and not unique to this application. Could you cite some other examples to demonstrate the challenge.

Yes, we have updated the text as follows:
> *So rather than the runoff flowing through a simple 1D river channel as mizuRoute simulates, much of the water stays in the wetlands, attenuating the flow. This is a common problem with river routing models, requiring an approach to represent floodplain storage or otherwise parametrise wetland processes (Zhao et al., 2017; Dadson et al., 2010; Fleischmann et al., 2018).*

Line 465 please specify why this area of the catchment was selected to demonstrate the inundation modelling
We have added the following explanation:
> *This is a section of the Brahmaputra river along with a number of tributaries. This includes the entirety of the Brahmaputra river (Jamuna) within Bangladesh, until 20km before its confluence with the Ganges (Padma) river. Furthermore, this region provides a good example of a flood captured during 2017 in the Sentinel-1 observations.*

Line 509 what would these limitations mean if this model cascade was being used to assess climate change impacts?
We have added the following text to address this:
> *The uncertainties in the model outputs mean that when using this model cascade, for example to assess climate change impacts, it would not be appropriate to focus on changes for individual pixels. Rather, changes aggregated to river reach or sub-catchment will be more robust.*

Line 547. The terminology used in the conclusion is slightly confusing / inconsistent i.e. "high resolution flood inundation", "fine scale flood risk information". These terms are relative, what is presented in this manuscript could be described as high resolution compared to what is currently available for flood modelling globally at these type of scales, but it is not high resolution in the more general context. The discussion within the manuscript has previously focussed on the challenges of adequately describing the topography, river network etc in the absence of high resolution data.

We take your point that these scales are relative. For the purposes of this study, we have used the term high resolution to refer to scales of less than 1km, so have updated the text to make this clearer.

*This paper presents a methodology for global modeling of high resolution flood inundation. The use of flood risk information at scales less than 1 km has the potential to transform how climate change risks relating to flooding are presented.*

Line 545 missing or incorrect word (the = this?)?
Thank you, we have corrected this.

Line ~550 would this model cascade be of potential use for flood forecasting in data/resource poor regions of the world or are the run times prohibitive?

Yes, we have added the following text:
*This methodology has the potential for being used for climate change research or for flood forecasting (for regions where high quality hydrological data-sets do not exists).*

Table 4 The definition of pluvial flooding refers to "flash flooding caused by short, very intense precipitation events". Could you clarify if you consider this to be before water enters a watercourse or if you would also include flooding from small watercourses. I think there is potential for confusion in the definition here as the methodology only includes watercourses with catchments >250m2. Including pluvial flooding would require a continuum of approaches to go from catchments of 250m2 to pluvial flooding which many not be as simple as implied in line 545.

You are correct that this is a continuum/ grey area about whether flooding of small watercourses is considered fluvial or pluvial. We've updated the pluvial definition in the table to be:
*Flash flooding caused by short very intense precipitation events. This may include flooding of streams and small rivers not resolved by the fluvial model*

Similarly, we've updated the related text:
*Pluvial flooding in particular should be a relatively simple addition to this framework although pluvial flooding occurs at smaller scales, this would require both higher resolution DEMs and precipitation than fluvial flooding. Including pluvial flooding may also need to be done in coordination with simulating fluvial flooding in smaller watercourses.*

**Anonymous Referee #2**

As a person who has been working with all the three components featured in this paper, i.e., meteorology, hydrologic modeling, and flood inundation mapping, I am seriously confused about this flood-cascade. I thoroughly read the paper. Overall, it is very well- written. The methodology is reasonable. Yet, I think that the authors' justification of creating a flood-cascade sends a wrong message to the new-generation of readers, disavowing the long history of hydrologic-hydraulic modeling research (apologies for the use of strong words here).

After reading the abstract and introduction, a reader might think that this "flood-cascade" is a brand-new concept. But that would be wrong. In fact, the so-called flood-cascade, integrating meteorology drivers to flood hazard predictions and hence assessment of climate change impacts, has existed in our scientific community for a long time with different forms, scales, and names. For example, the GLOFRIM for integrated hydrological–hydrodynamic global modelling by Hoch et al. (2017) and the National Water Model in the United States are just two examples of many existing, globally applicable flood-cascade frameworks. Each of these existing frameworks are well-resolved flood-cascades with cascading input-output structures according to the authors' definition. I hereby strongly oppose the author's narrative in the existing version of the paper.

In summary, the merit of this paper comes down to evaluation of a relatively new LISFLOOD modeling framework for one of the world's data-poor flood-prone basins. The paper has all the potential to getting accepted for publication, however, with major changes in the title, abstract, and introduction.

Thank you for your frank review of our manuscript. We take the major point of this review, that the concept of a model cascade for flood inundation is not new, and there are previous examples of this. We have taken this onboard and modified the abstract and introduction, to emphasise previous research that has been done in this area and reframed the motivation to emphasise that one of the key goals of this framework has been to make flood inundation modelling for climate change studies more accessible.

For example the text in the abstract:
> *Current climate change assessments of flood risk typically neglect key processes, and instead of explicitly modeling flood inundation, they commonly use precipitation or river flow as proxies for flood hazard. This is due to the complexity and uncertainties of model cascades and the computational cost of flood inundation modeling. Here, we lay out a clear methodology for taking meteorological drivers, e.g., from observations or climate models, through to high-resolution (~90 m) river flooding (fluvial) hazards. Thus, this framework is designed to be an accessible, computationally efficient tool using freely available data, to enable greater uptake of this type of modeling.*

Additionally, the second paragraph in the updated introduction is as follows:
> *The hydrological cycle is being altered due to the influences of climate change. So making flood inundation models more accessible for use in climate change impacts studies is an important step for future research. The approach presented here is designed to enable more robust flood inundation estimates using climate model outputs, over larger regions. This study focuses on describing the model framework and evaluating its performance against past observations of river flow and flood inundation. The model framework is compatible with climate model outputs, which use the same data format as gridded meteorological data. However, we do not include climate projections here, which will be covered in a future study.*